# Opposing roles for striatonigral and striatopallidal neurons in dorsolateral striatum in consolidating new instrumental actions

Alexander C. W. Smith [1,6], Sietse Jonkman[1,6], Alexandra G. Difeliceantonio [1,5], Richard M. O'Connor[1], Soham Ghoshal [1,2], Michael F. Romano[3], Barry J. Everitt [4] & Paul J. Kenny [1✉]

Comparatively little is known about how new instrumental actions are encoded in the brain. Using whole-brain c-Fos mapping, we show that neural activity is increased in the anterior dorsolateral striatum (aDLS) of mice that successfully learn a new lever-press response to earn food rewards. Post-learning chemogenetic inhibition of aDLS disrupts consolidation of the new instrumental response. Similarly, post-learning infusion of the protein synthesis inhibitor anisomycin into the aDLS disrupts consolidation of the new response. Activity of D1 receptor-expressing medium spiny neurons (D1-MSNs) increases and D2-MSNs activity decreases in the aDLS during consolidation. Chemogenetic inhibition of D1-MSNs in aDLS disrupts the consolidation process whereas D2-MSN inhibition strengthens consolidation but blocks the expression of previously learned habit-like responses. These findings suggest that D1-MSNs in the aDLS encode new instrumental actions whereas D2-MSNs oppose this new learning and instead promote expression of habitual actions.

[1] Nash Family Department of Neuroscience, Icahn School of Medicine at Mount Sinai, New York, NY, USA. [2] Hunter College, City University of New York, New York, NY, USA. [3] Department of Computational Neuroscience, Boston University, Boston, MA, USA. [4] Department of Psychology, University of Cambridge, Cambridge, UK. [5] Present address: Department of Human Nutrition, Foods and Exercise, College of Agriculture and Life Sciences and Center for Transformative Research on Health Behaviors, Fralin Biomedical Research Institute, Virginia Tech, VA, USA. [6] These authors contributed equally: Alexander C. W. Smith, Sietse Jonkman. ✉email: paul.kenny@mssm.edu

L earning new actions that deliver beneficial outcomes is a fundamental form of behavioral plasticity and a key function of the brain[1,2]. New instrumental learning occurs when an action unexpectedly results in delivery of a rewarding stimulus or the withdrawal of a punishing stimulus, thereby reinforcing the behavior by increasing the likelihood that the same action will be executed again under similar circumstances[1]. This new learning involves the consolidation of neural signals that represent the sequence of motor behaviors and associated sensory, interoceptive, and spatiotemporal information that preceded receipt of the unexpectedly beneficial outcome[3]. Such diverse neural signals are known to converge in dorsal and ventral regions of the striatum[4,5], a brain structure considered critical in linking motor actions to motivational states[6]. As such, the striatum is hypothesized to play a major role in the consolidation processes by which newly learned instrumental actions are encoded in the brain[7,8].

Surprisingly little is known about how new instrumental actions are encoded by the striatum. The nucleus accumbens (NAc) region of ventral striatum is thought to track and evaluate the outcome of recently executed actions to detect unexpected outcomes, so-called reward prediction errors, to refine future actions[9,10]. By detecting reward prediction errors generated by actions that yield unexpectedly beneficial outcomes, the NAc has long been viewed as the striatal region most likely to regulate new instrumental learning[7,10]. Recent findings have challenged this concept and raised questions about the precise nature of NAc involvement in this process[8,11]. Currently, there is little evidence to support a role for the dorsolateral region of the striatum (DLS) in new instrumental learning[9,10,12]. Instead, the DLS is considered a core component of the brain's habit system that regulates the expression of previously learned instrumental responses independent of any representation of the reward that originally reinforced that response[13–15]. Specifically, the aDLS is thought to specialize in stimulus-response learning in which conditioned stimuli in the environment come to elicit a previously learned instrumental response, with such value-independent sensorimotor behavior considered critical for the development and persistence of habitual actions[16,17]. However, new instrumental learning often coincides with new sensorimotor learning, leading to speculation that some of the same brain structures may participate in both types of learning[18]. Here, we report findings from a comprehensive series of experiments designed to identify regions of the striatum involved in consolidating new instrumental actions and to explore the cellular mechanism involved in this process. We provide evidence that the aDLS and NAc act in a coordinated fashion to consolidate dissociable aspects of newly learned instrumental actions and that striatonigral and striatopallidal neurons in the aDLS act in a functionally antagonistic manner to control the consolidation process.

## Results

**New instrumental learning restructures behavior**. To isolate the processes involved in new instrumental learning at the earliest stages, we used a behavioral task in which animals were permitted to acquire a new lever press response for food rewards under a continuous reinforcement (fixed ratio 1; FR1) schedule, in the absence of any conditioned stimuli, during a single training session[19] (Fig. 1a; see Methods). We found that hungry rats permitted to lever-press for food pellets until they earned a total of 50 rewards (50:0 rats) executed the same response vigorously when tested 48 h later under extinction conditions (retention test; 60 min session) (Fig. 1b and Supplementary Fig. 1). Rats that received 50 pellets non-contingently during the acquisition session (0:50 rats), when lever presses had no scheduled consequence, responded at low rates during the retention

test (Fig. 1b). Rats permitted to lever-press for only 10 pellets during the acquisition session, after which the lever was retracted and 40 pellets were delivered non-contingently (by yoking delivery to a 50:0 rat) to control for arousal associated with pellet delivery (10:40 rats), also responded at low rates during the retention test (Fig. 1b and Supplementary Fig. 1). However, the 10:40 rats tended to have higher rates of lever-pressing than 0:50 rats during the retention test (Fig. 1b), although this effect failed to reach statistical significance. This suggests that subthreshold learning may have occurred in 10:40 rats triggered by successfully earning 10 response-contingent food rewards early in the training session, with this nascent learning weakened by the subsequent withdrawal of the lever and delivery of 40 rewards in a non-contingent manner. These data suggest that the 50:0 rats, but not the 0:50 or 10:40 rats, reliably acquire a new instrumental action during a single training session.

Closer inspection of the data collected during the retention test revealed two discrete patterns of responding: either solitary lever presses or engagement bouts of lever presses emitted in rapid succession (<5 s apart; Supplementary Fig. 2). 50:0 rats engaged far more frequently in bouts of responding (Fig. 1c) than 10:40 or 0:50 rats during the retention test, whereas the numbers of presses per response bout (i.e., bout density) were similar between all three groups (Fig. 1d). Numbers of solitary lever presses were also modestly increased in 50:0 rats compared with 10:40 and 0:50 rats (Supplementary Fig. 3). It has been reported that rats executing a previously learned instrumental response for food rewards demonstrate similar bouts of responding interspersed by solitary responses[20–23], with bout frequency influenced by the strength/intensity of the reinforcer and bout density influenced by the effort required to obtain the reinforcer[24–27]. In 50:0 rats, bout frequency but not bout density or solitary responses was correlated with numbers of magazine entries during the retention test (Supplementary Fig. 3). This suggests that engaging in bouts of responding was closely linked to reward retrieval behaviors, consistent with this feature of behavior reflecting the animals successfully encoding the relationship between the new instrumental action and reward delivery.

Next, we confirmed that mice can also learn a new lever-press response during a single training session, similar to rats. This was important because it would enable us to utilize behavioral genetics tools available in mice but not currently available or optimized in rats (see below). Hungry C57BL6/J mice learned to lever-press under an FR1 schedule to earn 30 food pellets during a single training session, using the same procedure described above for 50:0 rats (Supplementary Fig. 4). These mice responded vigorously during a retention test 48 h later (Supplementary Fig. 4), suggesting that they had successfully encoded the new instrumental response. Notably, the mice demonstrated bouts of lever-pressing interspersed by solitary lever-presses during the retention test similar to 50:0 rats (Supplementary Fig. 4). This prompted us to investigate the relationship more thoroughly between instrumental contingencies during the training session and the subsequent expression of response bouts during the retention test. Specifically, we trained a group of mice to lever-press for 30 chow pellets under a FR1 schedule (FR1-Chow mice), as described above. A second group of mice was trained to lever-press for palatable sucrose pellets under an FR1 schedule during the training session (FR1-Sucrose mice). In this manner, we could investigate the effects of manipulating reward magnitude during training on the microstructure of responding during the retention test. A third group of mice responded for chow pellets under an FR2 schedule during training (FR2-Chow mice), allowing us to investigate the effects of increasing the effort required to earn each reinforcer during training on subsequent bout structure.

FR1-Sucrose mice responded far more vigorously than FR1-Chow mice during the retention test (Supplementary Fig. 5).

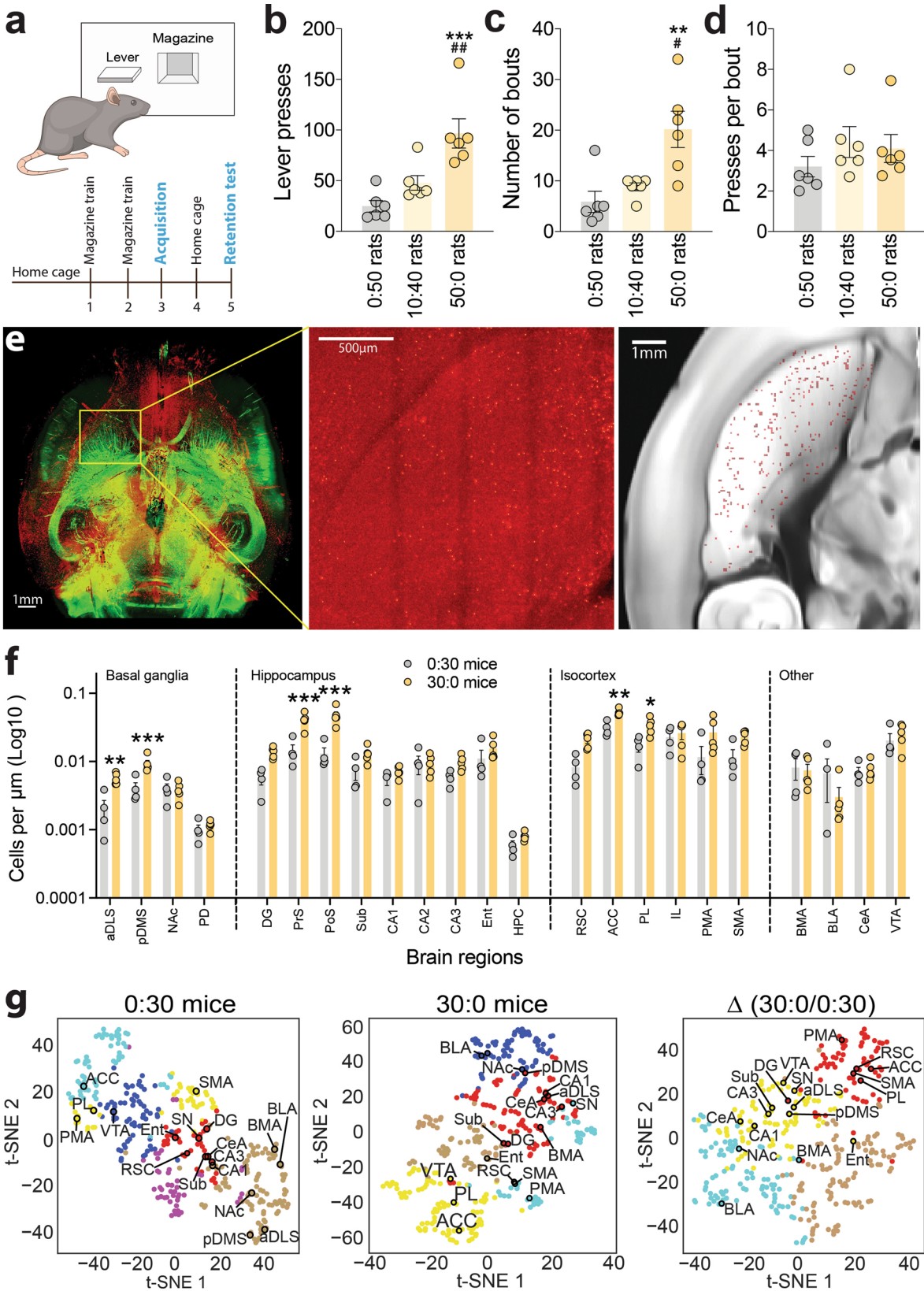

Numbers of lever presses per response bout (bout density), and numbers of solitary lever-presses were similar in FR1-Sucrose and FR1-Chow mice (Supplementary Fig. 5). By contrast, the frequency of response bouts was much higher in FR1-Sucrose than FR1-Chow mice (Supplementary Fig. 5). FR2-Chow mice also responded more vigorously than FR1-Chow mice during the retention test (Supplementary Fig. 5). There was a trend for increased bout frequency in FR2-Chow mice compared with FR1-Chow mice, which did not achieve statistical significance (Supplementary Fig. 5). However, bout density was markedly increased in FR2-Chow mice compared with FR1-Chow mice (Supplementary Fig. 5). These data suggest that that bout

**Fig. 1 Instrumental learning recruits neural activity in the aDLS. a** Graphical representation of task used to investigate mechanisms of new instrumental learning and experimental design. **b** Mean (±s.e.m.) number of lever presses in 50:0 rats ($n = 6$), 10:40 rats ($n = 6$) and 0:50 rats ($n = 6$) during retention test; $F_{(2,15)} = 13.8$, $p = 0.0004$, One-way ANOVA; ***$P = 0.0004$ compared with 0:50 rats, ##$p = 0.0097$ compared with 10:40 rats, post-hoc test. **c** Mean (±s.e.m.) number of bouts of lever presses; $F_{(2,15)} = 9.587$, $p = 0.0021$; **$P = 0.0004$ compared with 0:50 rats, #$p = 0.0097$ compared with 10:40 rats, post-hoc test. **d** Mean (±s.e.m.) number of lever presses per bout; $F_{(2,15)} = 0.8937$, $p = 0.4299$. **e** iDISCO-cleared brain and whole-brain c-Fos detection from 30:0 and 0:30 mice (representative image; left panel), were processed through the ClearMap Python package to map c-Fos+ cells to discrete brain regions (middle panel), then processed through custom Python scripts to determine c-Fos+ cell counts across striatal subregions (right panel). **f** Mean (±s.e.m.) number of c-Fos+ cells in basal ganglia, hippocampal, isocortex and other brain regions relevant to learning and motivation were compared in 0:30 and 30:0 mice. *$P < 0.05$, **$p < 0.01$, ***$p < 0.001$ compared with 0:30 mice, post-hoc test after significant interaction effect in Two-way ANOVA. Full names of abbreviated brain structures are provided in Supplementary Table 1. **g** KNN and tSNE analyses were used to cluster brain structures based on whole-brain patterns of c-Fos+ cells in 0:30 mice (left panel), and 30:0 mice (middle panel), and the differences in c-Fos+ cells between 0:30 and 30:0 mice (right panel).

frequency during the retention test was more sensitive than bout density to changes in the relative value of the reinforcer earned during the training session. Conversely, bout density was more sensitive than bout frequency to changes in the effort required to earn the reinforcer during training.

**Anterior dorsolateral striatum activity is modified by new instrumental learning.** Next, we used whole-brain clearing and c-Fos mapping to identify brain regions recruited by new instrumental learning in an unbiased manner using the iDISCO+ brain clearing procedure[28]. Because of the availability of mouse brain atlases suitable for registration of light-sheet microscopy images[29,30], mice were used for this experiment. A new group of 30:0 mice were trained to acquire the lever-press response during a single session then killed 60 min afterwards. For comparison, brains were collected from control mice that received 30 pellets non-contingently during the training session (0:30 mice). Intact brains from both groups were processed using the iDISCO+ procedure and immunostained for c-Fos expression[28]. The ClearMap Python package was used to detect and register c-Fos immuno-positive (c-Fos+) cells onto the Allen Brain Atlas to map brain-wide patterns of c-Fos expression, and custom Python scripts were used to parse cell counts across striatal subregions[31] (Fig. 1e). Using this approach, we found that numbers of c-Fos+ cells were increased in cortical, limbic, basal ganglia, midbrain and hindbrain sites in 30:0 mice compared with 0:30 mice (Table S1). In the striatum, c-Fos levels were increased in posterior dorsomedial (pDMS) and anterior dorsolateral (aDLS) regions of 30:0 mice compared with 0:30 mice (Fig. 1f) but were unaltered in the anterior DMS (aDMS), posterior DLS (pDLS) or NAc (Fig. 1f). We performed unbiased K-nearest neighbor (KNN) clustering and t-distributed stochastic neighborhood embedding (tSNE) analyses on density of c-Fos+ cells across the entire brains of 0:30 and 30:0 mice (Fig. 1g). The KNN analysis clustered together brain sites in which densities of c-Fos+ cells covaried across animals in the same groups, such that their closer proximity in the tSNE plot reflected greater covariance (Fig. 1g). This revealed that the aDLS and pDMS in 30:0 mice but not in 0:30 mice cluster with brain regions known to regulate learning and memory, such as components of the hippocampal complex and the amygdala (Fig. 1g). Further, KNN analysis of differences in the density of c-Fos expression between mice in the 30:0 and 0:30 groups showed that acquisition of the new lever-press response resulted in the aDLS and pDMS clustering with cortical and subcortical regions known to regulate the expression of instrumental actions (Fig. 1g), such as the ventral tegmental area and substantia nigra. These data suggest that new instrumental learning modifies neural activity in the pDMS, aDLS, and a broader network of cortical, hippocampal and basal ganglia brain regions involved in learning and motivation. To confirm these findings, immunohistochemical labeling of c-Fos was performed on striatal cryosections from groups of 50:0 and 0:50 rats

euthanized 60 min after their training session. Again, new instrumental learning was associated with increased numbers of c-Fos+ cells in the pDMS and aDLS, but not the aDMS, pDLS or NAc regions of the striatum (Supplementary Fig. 6). These data suggest that the pDMS and aDLS may participate in the consolidation of newly learned instrumental actions.

**Anterior dorsolateral striatum consolidates new instrumental learning.** To define regions of the striatum involved in consolidating new instrumental learning, we examined the effects of blocking protein synthesis in striatal subregions of rats soon after instrumental learning on the retention of the new response. We used rats for this experiment because the effects of striatal infusions of the protein synthesis blocker anisomycin on new instrumental learning have been previously reported in rats but not mice[7,11]. Bilateral indwelling cannulae were surgically implanted above the pDMS, pDLS, aDLS, or NAc core of rats (Supplementary Fig. 7), which were then trained to acquire a new lever-press response for food rewards during a single session according to the same procedure described above. Anisomycin was infused into these striatal sites soon after the completion of the acquisition session, then retention of the new response was assessed 48 h later (Fig. 2a, b). Anisomycin triggers aversion to food items consumed soon before its infusion into the striatum of rats[11]. To avoid this confound, we protected the value of the chow pellets earned during the acquisition session by permitting rats to briefly (5 min) consume an alternative reinforcer (20% sucrose solution) immediately after training but just before anisomycin infusion[11] (Fig. 2a). Using this approach, anisomycin-induced food aversion was detected only when sucrose solution but not chow pellets were subsequently made available (Supplementary Fig. 8), confirming that anisomycin did not alter the valuation of the earned chow pellets in any group of rats. The pDMS showed increased c-Fos expression after new instrumental learning (Fig. 1f) and is known to regulate the expression of previously learned instrumental actions[32], consistent with a role for the pDMS in consolidating new instrumental actions. We were therefore surprised that post-acquisition infusion of anisomycin into the pDMS did not alter any aspect of responding during the subsequent retention test compared with vehicle-infused rats (Supplementary Fig. 9). Similar to the pDMS, post-learning infusion of anisomycin into the pDLS did not alter responding during the retention test (Supplementary Fig. 9). By contrast, anisomycin infused into the aDLS soon after acquisition (Fig. 2c), but not 6 h later when the consolidation process is expected to be complete, reduced lever-pressing during the retention test (Fig. 2d). This finding was unexpected considering the well-established role of the aDLS in regulating previously reinforced behaviors that have become habitual in nature, and are expressed independent of any representation of the reward that originally reinforced that response[15]. Notably, these rats demonstrated a

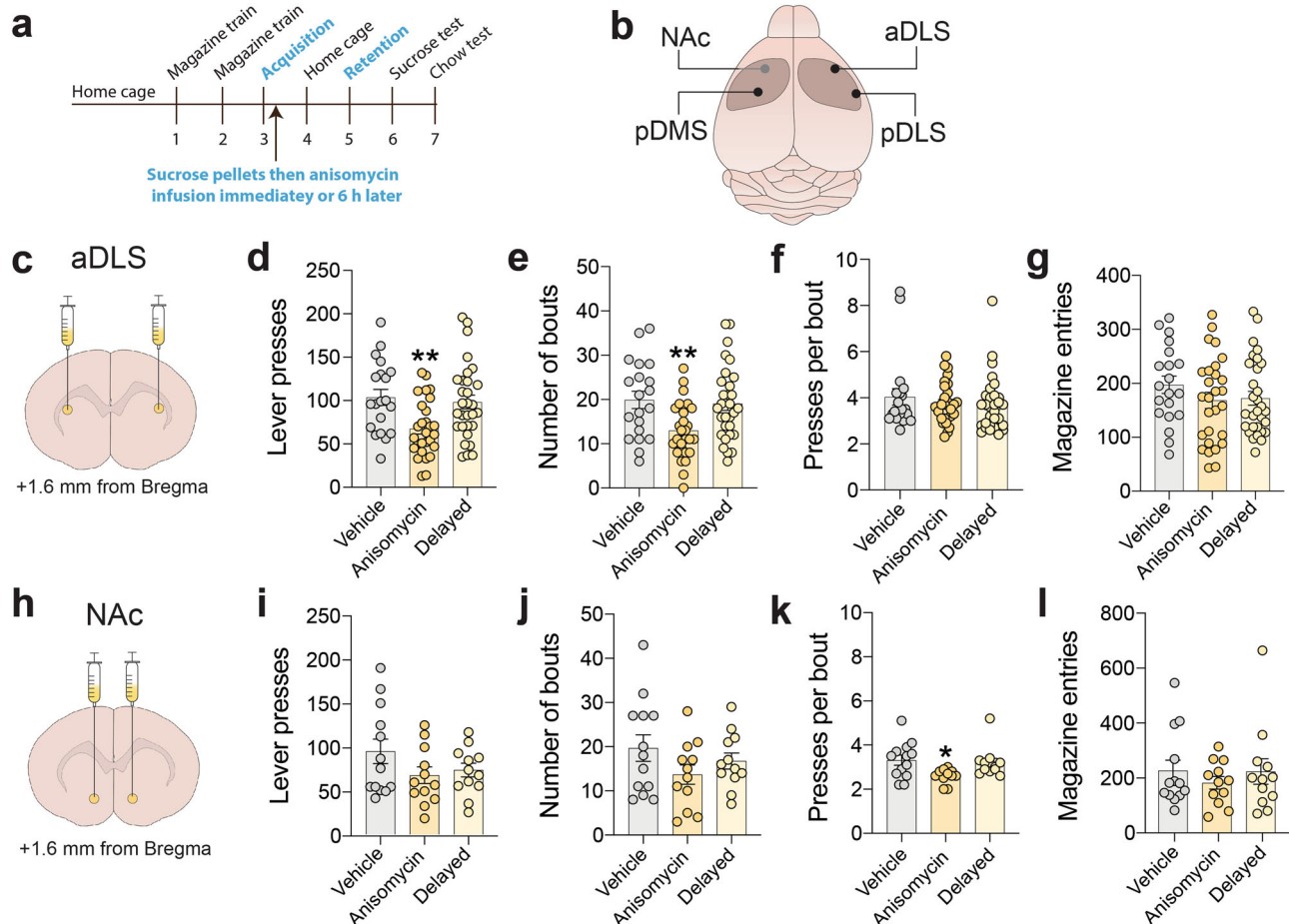

**Fig. 2 Protein synthesis in the aDLS is required for consolidation of new learning. a** Graphical representation of experimental design. **b** Regions of striatum targeted for post-learning anisomycin infusion. **c** Region of aDLS targeted with anisomycin. **d** Mean (±s.e.m.) number of lever presses in vehicle-treated rats ($n = 20$), rats infused with anisomycin into aDLS soon after new learning ($n = 28$), and rats infused with anisomycin 6 h after new learning ($n = 30$); $F_{(2,75)} = 6.442$, $p = 0.0026$, One-way ANOVA; $**P = 0.0047$ compared with vehicle-treated rats, post-hoc test. **e** Mean (±s.e.m.) number of bouts of lever presses (±s.e.m.); $F_{(2,75)} = 6.258$, $p = 0.0031$; $**P = 0.0062$ compared with vehicle-treated rats, post-hoc test. **f** Mean (±s.e.m.) number of lever presses per bout; $F_{(2,75)} = 0.4625$, $p = 0.6315$. **g** Mean (±s.e.m.) number of magazine entries; $F_{(2,\ 75)} = 0.9933$, $p = 0.3752$. **h** Region of NAc targeted with anisomycin. **i** Mean (±s.e.m.) number of lever presses in vehicle-treated rats ($n = 13$), rats infused with anisomycin into NAc soon after new learning ($n = 12$), and rats infused with anisomycin 6 h after new learning ($n = 12$); $F_{(2,\ 34)} = 1.744$, $p = 0.1901$, One-way ANOVA. **j** Mean (±s.e.m.) number of bouts of lever presses (±s.e.m.); $F_{(2,34)} = 1.552$, $p = 0.2264$. **k** Mean (±s.e.m.) lever presses per bout; $F_{(2,34)} = 4.341$, $p = 0.0209$; $*P = 0.0181$ compared with vehicle-treated rats, post-hoc test. **l** Mean (±s.e.m.) number of magazine entries; $F_{(2,34)} = 0.4294$, $p = 0.6543$.

decrease in the number of response bouts (Fig. 2e), but not bout density (Fig. 2f) or the number of solitary responses (Supplementary Fig. 10), consistent with frequency of engagement bouts being the feature of behavior that best corresponds with new instrumental learning. The number of head-entries into the food magazine was unaltered during the retention test in these rats (Fig. 2g), suggesting that only behaviors directed toward executing the new lever-press action, but not those involved in reward retrieval, were impacted by post-learning disruption of new protein synthesis in the aDLS. Finally, post-acquisition infusion of anisomycin into the NAc (Fig. 2h) tended to reduce lever-pressing during the retention test (Fig. 2i). There was also trend for reduced bout frequency in these animals. This suggests that increasing the group size may have yielded statistically significant effects of anisomycin on overall lever-pressing and bout frequency. However, the density of response bouts was decreased (Fig. 2k), but not bout frequency (Fig. 2j), magazine entries (Fig. 2l) or solitary lever presses (Supplementary Fig. 10), during the retention test in these rats. The fact that bout density was robustly decreased in these animals suggests that this feature of behavior is particularly sensitive to NAc manipulations. These

data suggest that the aDLS plays an important role in consolidating new instrumental actions and that the aDLS and NAc act in a cooperative manner to preferentially encode the relative value of the reward delivered by a new action and the effort required to earn that reward, respectively.

To confirm the role of the aDLS in new instrumental learning, we chemogenetically silenced this site in mice immediately after they acquired a new lever-press response for food rewards, then assessed their performance in a retention test 48 h later. Specifically, C57Bl/6J mice were injected with AAV8-hSyn-hM4Di-mCitrine into the aDLS and we confirmed that virus-infected cells were detected only in the aDLS (Supplementary Fig. 11). We trained the mice to lever press for 30 food pellets in a single training session, then injected them with vehicle or clozapine-N-oxide (CNO; 3 mg kg$^{-1}$) immediately after the acquisition session or 6 h later (Fig. 3a). Consolidation of the new response was disrupted by immediate but not delayed post-learning injection of CNO, reflected by decreased lever-pressing compared with vehicle-treated mice during the retention test (Fig. 3b). Once again, bout frequency (Fig. 3c), but not bout density (Fig. 3d) or the number of solitary responses

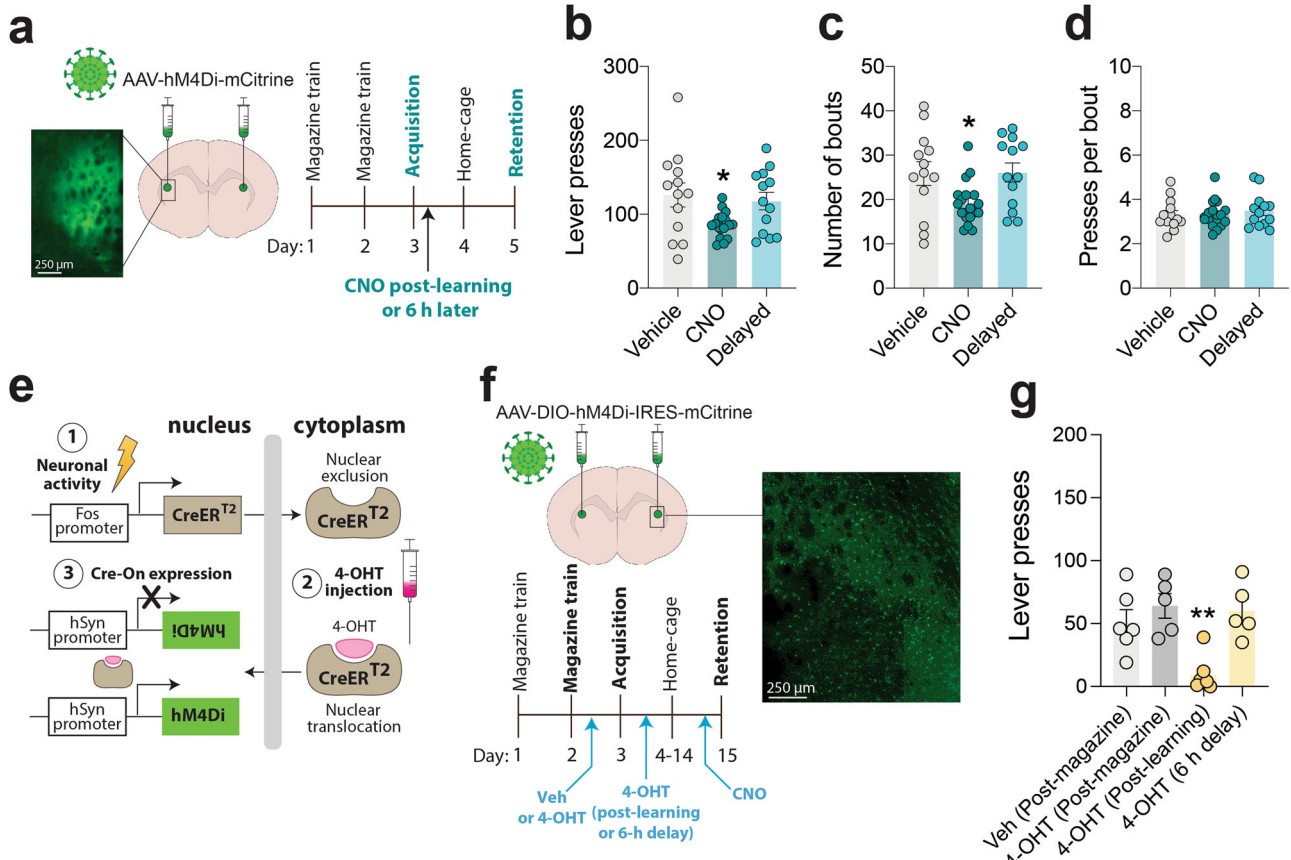

**Fig. 3 aDLS activity regulates consolidation of new learning. a** Graphical representation of inhibitory hM4Di-mCitrine DREADD virus delivered to aDLS of mice, experimental design, and representative mCitrine image from aDLS-injected mouse. **b** Mean (±s.e.m.) number of lever presses in vehicle-treated mice ($n = 13$), mice injected with CNO after new learning ($n = 18$), and mice injected with CNO 6 h after new learning ($n = 13$); $F_{(2,42)} = 4.346$, $p = 0.0194$, One-way ANOVA; *$P = 0.0247$ compared with vehicle-treated mice, post-hoc test. **c**, Mean (±s.e.m.) number of bouts of lever presses; $F_{(2,41)} = 4.536$, $p = 0.0166$; *$P = 0.0434$ compared with vehicle-treated rats, post-hoc test. **d** Mean (±s.e.m.) number of lever presses per bout; $F_{(2,41)} = 0.2868$, $p = 0.7522$. **e** Graphical representation of c-Fos-stimulated CreER[T2] transcription (1), 4-OHT-mediated nuclear translocation of CreER[T2] (2), and CreER[T2]-mediated gene expression (3). **f** Graphical representation of inhibitory Cre-dependent DIO-hM4Di-mCherry DREADD virus delivered to aDLS of *Fos-CreER[T2]* mice, experimental design, and representative image of mCitrine expression in aDLS of *Fos-CreER[T2]* mice. *Fos-CreER[T2]* mice were injected with vehicle (Veh; $n = 6$) or 4-OHT ($n = 5$) after magazine training or injected with 4-OHT immediately ($n = 7$) or 6 h ($n = 5$) after the acquisition session. **g** Mean (±s.e.m.) number of lever presses in the four groups of CNO-treated *Fos[2A-iCreERT2]* mice during the retention test; $F_{(3, 19)} = 9.56$, $p = 0.0005$; **$P < 0.01$, compared with mice treated with vehicle after magazine training, post-hoc test.

(Supplementary Fig. 12), was the feature of behavior that was most impacted, further suggesting a role for the aDLS in encoding the value of the new action. Consumption of a sucrose solution (20%) was similar in hM4Di-mCitrine mice after vehicle or CNO (3 mg kg$^{-1}$) injection (Supplementary Fig. 13). Exploratory behavior in an open field was also similar in hM4Di-mCitrine mice after vehicle or CNO injection (Supplementary Fig. 13). Hence, abnormalities in taste processing or motor behavior are unlikely to account for the post-learning deficit in consolidation detected in these mice. We also found that the numbers and patterns of lever presses during a retention test were similar in vehicle- and CNO-treated control mice that had been injected with an AAV8-hSyn-GFP virus into the aDLS (Supplementary Fig. 14), suggesting that CNO alone did not alter new instrumental learning.

Next, we investigated the effects of selectively inhibiting only those aDLS neurons active during the consolidation of a new instrumental response on the subsequent expression of the response. We infused AAV8-hSyn-DIO-HA-hM4Di-IRES-mCitrine into the aDLS of *Fos[2A-iCreERT2]* mice[33]. In these mice, CreER[T2] is expressed from the *Fos* gene locus in an activity-dependent manner and is retained in the cytoplasm (Fig. 3e). CreER[T2] can translocate to the nucleus for

a short time period after treatment with tamoxifen (~12 h window) or its derivative 4-hydroxytamoxifen (4-OHT; ~4 h window), where it can drive Cre-dependent gene expression in a spatiotemporally controlled manner, a process termed "targeted recombination in active populations" (TRAP) (Fig. 3e)[33]. After DIO-hM4Di-IRES-mCitrine injection, *Fos[2A-iCreERT2]* mice were trained in the same lever-press procedure described above (Fig. 3f). We injected two groups of the *Fos[2A-iCreERT2]* mice with either vehicle or 4-OHT immediately after magazine training to trigger nuclear translocation of Cre and thereby drive hM4Di expression in those aDLS neurons that were active immediately after magazine training (Fig. 3f). Next, we trained all *Fos[2A-iCreERT2]* mice to lever-press for 30 food pellets during a single training session, then injected the remaining untreated mice with 4-OHT immediately after training or 6 h later. In this manner, we could drive hM4Di expression in aDLS neurons active during the post-learning consolidation period or those active 6 h later. Finally, we treated all 4 groups of *Fos[2A-iCreERT2]* mice with CNO (3 mg kg$^{-1}$) 10 min before a retention test conducted 48 h later. We found that CNO decreased lever-pressing during the retention test in *Fos[2A-iCreERT2]* mice that received 4-OHT immediately after new instrumental conditioning but had no effects in the other three groups of *Fos[2A-iCreERT2]* mice (Fig. 3g). This

suggests that inhibiting neuronal ensembles in aDLS engaged during the consolidation of a new instrumental action, but not those ensembles active at other time-points, disrupts the consolidation of the newly learned action. Next, we examined the effects of selectively stimulating aDLS ensembles during consolidation on the subsequent expression of the new lever-press response. Specifically, we injected a group of Fos[CreERT2] mice with AAV8-hSyn-DIO-hM3Dq-mCherry into the aDLS, then injected these mice with vehicle or 4-OHT immediately after the training session (Supplementary Fig. 15). Post-learning CNO treatment increased the expression of the new lever-press response during a retention test in the 4-OHT-treated but not vehicle-treated Fos[CreERT2] mice (Supplementary Fig. 15), suggesting that artificially stimulating those neuronal ensembles in aDLS that were active during the post-learning period facilitated the consolidation process. Collectively, these findings support a key role for the aDLS in consolidating newly learned instrumental actions.

**Striatonigral MSNs in dorsolateral striatum consolidate new instrumental actions.** Between 90 and 95% of neurons in the striatum are GABAergic medium spiny neurons (MSNs) that are generally divided into two categories based on their dopamine receptor expression and projection profiles. MSNs that express dopamine D1 receptors comprise a direct pathway from the basal ganglia (striatonigral cells), whereas MSNs that express D2 receptors comprise an indirect pathway (striatopallidal cells). D1 and D2-MSNs often have opposing roles in motor control, reward, and motivation[34–36], but less is known about their contributions to striatal-dependent learning processes. To investigate the potential involvement of D1 and D2-MSNs in new instrumental learning, we assessed changes in the morphology of dendritic spines expressed by MSNs in the aDLS as a marker of learning-related structural plasticity. We injected AAV-DIO-GFP into the aDLS of D1-Cre mice to selectively express GFP in D1-MSNs. Then, after recovery, mice were permitted to lever-press for 30 food rewards (30:0 mice), or received 30 pellets non-contingently (0:30 mice), and were killed 15 min after the session (Fig. 4a). Spines on MSNs in the aDLS from these animals were identified using DiOlistic labeling with tungsten particles coated with the lipophilic carbocyanine dye DiI (Fig. 4a–d). We found that spine head diameter (Fig. 4e,g), but not spine density (Fig. 4f), was increased in GFP+ (presumptive D1) MSNs but not in GFP- (presumptive D2) MSNs in 30:0 mice compared with 0:30 mice. This suggests that D1-MSNs in aDLS undergo structural remodeling in response to new instrumental learning.

It was unclear if the learning-related structural plasticity in D1-MSNs reflected their involvement in the consolidation process or their more generalized recruitment during the training session when mice expressed the new instrumental response for the first time. Therefore, to better understand the cellular mechanisms of consolidation, we used in vivo calcium imaging to monitor the activity of D1 and D2 cells in the aDLS during the critical period immediately after new instrumental learning. To accomplish this, we injected AAV8-hSyn-DIO-GCaMP6m into the aDLS of D1-Cre or D2-Cre mice to express the genetically encoded calcium indicator GCaMP6m, then implanted GRIN lenses above sites of virus infection (Fig. 4h). After recovery, mice were habituated to the attachment of head-mounted microendoscopes (miniscopes) for 7 days[37], then permitted to lever-press for 30 food rewards (30:0 mice), according to the same procedure described above (Supplementary Fig. 16). D1 and D2 MSN activity was monitored for 15 min immediately after the acquisition session, when consolidation is sensitive to blockade of new protein synthesis or chemogenetic inhibition. For comparison, we also recorded neural activity immediately after the second magazine training session in these same animals, when food pellets were delivered

noncontingently (Fig. 4a). A second group of D1-Cre::DIO-GCaMP6m and D2-Cre::DIO-GCaMP6m mice were permitted to lever-press for only 10 pellets during the acquisition session, then received 20 pellets noncontingently (10:20 mice). We similarly recorded D1 and D2 MSN activity in these control mice immediately after magazine training and after the acquisition session. As expected, the 30:0 mice responded vigorously during the retention test 48 h later, suggesting that they had successfully encoded the new instrumental action, whereas the 10:20 mice responded far less vigorously (Supplementary Fig. 16). We detected a robust increase in D1 MSN activity in the 30:0 mice during the post-learning consolidation period compared with their activity after magazine training (Fig. 4i). This is consistent with recent data suggesting that increases in D1 MSN activity throughout the entire dorsal striatum correlate with new instrumental leaning in mice when learning occurs across multiple training sessions[21]. Conversely, D2-MSNs showed a striking decrease in activity in the 30:0 mice during this same period (Fig. 4j).

In 10:20 mice, we did not detect any changes in D1 MSN activity during the post-acquisition period compared with the post-magazine training (baseline) period (Fig. 4i), which did not reliably encode the new instrumental response compared with 30:0 mice. However, D2 MSN activity was increased during the post-acquisition consolidation period compared with the baseline period in these mice (Fig. 4j). The fact that 10:20 mice showed increased D2 MSN activity during the post-consolidation period, which is opposite to the decreased D2 MSN activity in 30:0 mice during the same period, suggests that the failure of 10:20 mice to consolidate the new lever-press response may not be a passive process that reflects poor learning because of a limited number of training opportunities. Instead, this may reflect an active process in which D2 MSN activity is engaged during the consolidation phase to 'overwrite' a nascent instrumental response that was initially beneficial during the early stages of training session but then rendered obsolete by a change in the instrumental contingencies. Together, these findings suggest that consolidation of a new instrumental action is associated a dramatic shift in the balance of D1 and D2 MSN activity in the aDLS, with increased D1 MSN activity likely involved in consolidating the new response into long-term storage. Conversely, D2-MSNs may execute a quality control function, with post-learning decreases in their activity facilitating the consolidation of an advantageous new instrumental response and increases in their activity impeding the consolidation of non-beneficial action sequences.

The calcium imaging data described above suggest that D1-MSNs in aDLS encode new instrumental actions during the post-learning consolidation period and that a period of D2-MSN quiescence facilitates this process. To test these predictions, we investigated the effects of post-learning inhibition of D1 or D2-MSNs in the aDLS on consolidation of the new lever-press response. Specifically, we injected AAV8-hSyn-DIO-hM4Di-mCherry into the aDLS of D1-Cre and D2-Cre mice (Fig. 5a), waited 3 weeks, then trained them to lever-press for food pellets during a single session using the same procedures described above (Fig. 5b). The mice were then injected with vehicle or CNO (3 mg kg$^{-1}$) immediately after they acquired the new action or 6 h later, and their performance was assessed in a retention test under extinction conditions 48 h later (Fig. 5b). Strikingly, inhibition of D1-MSNs in the aDLS during the consolidation phase immediately after new learning, but not 6 h later, almost completely abolished the execution of the newly learned response during the subsequent retention test (Fig. 5c). Once again, the frequency of response bouts (Fig. 5d), but not bout density (Fig. 5e), was the feature of behavior most impacted. Conversely, chemogenetic inhibition of D2-MSNs in the aDLS soon after new learning, but

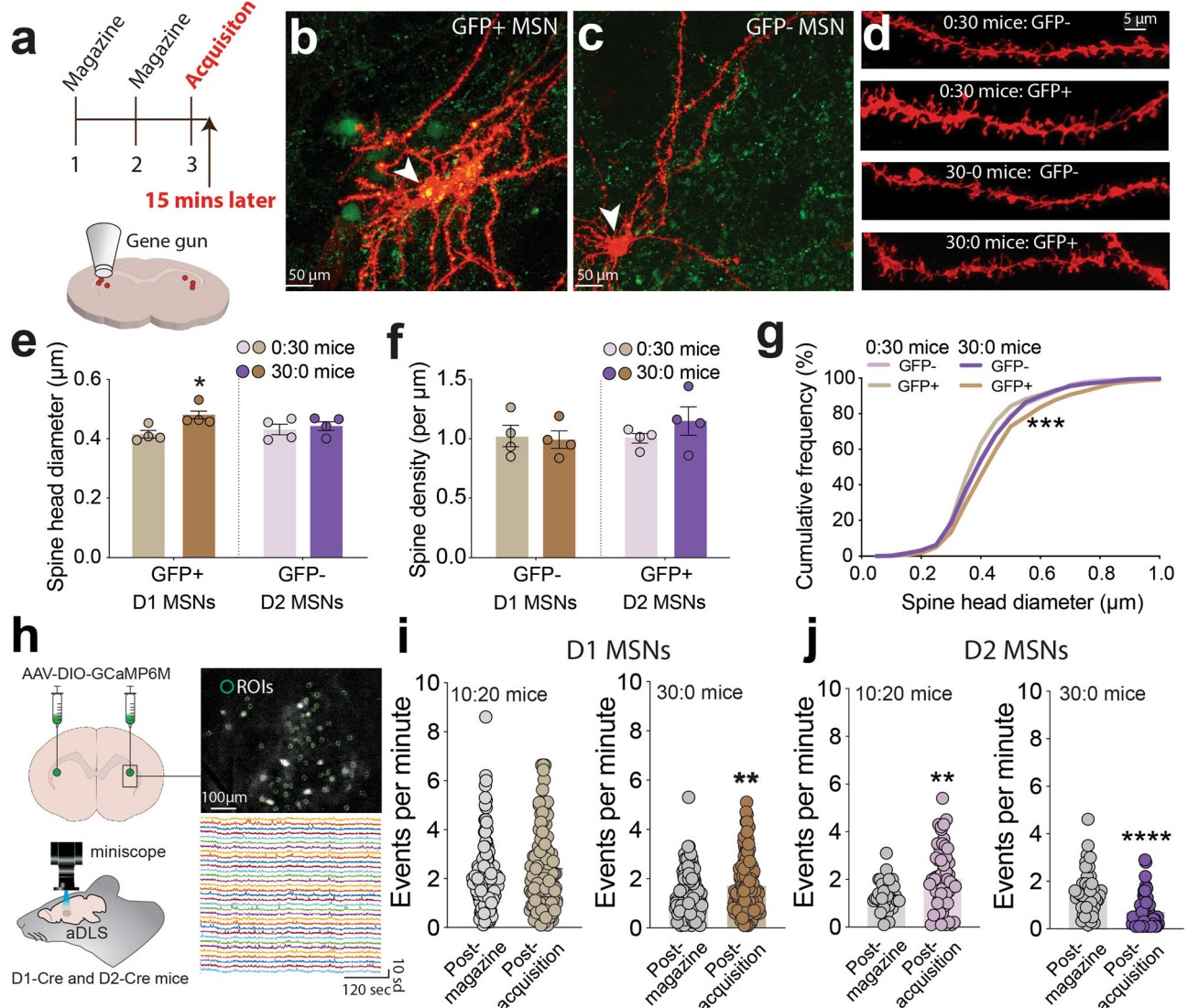

**Fig. 4 Opposite modulation of D1 and D2 MSN activity in aDLS during consolidation. a** Graphical representation of experimental design (upper) and the DiOlistic labeling of MSNs using DiI-coated tungsten particles (lower). **b** Representative image of a DiI-filled MSN (red) that expressed GFP (green; GFP+) in aDLS from D1-Cre mice injected with AAV-DIO-GFP (note yellow-colored cell body; presumptive D1 MSN). **c** Representative image of a DiI-filled MSN (red) that did not express GFP (GFP−; presumptive D2 MSN). **d** High-resolution confocal images of dendrites from DiI-labeled GFP+ and GFP- MSNs from the aDLS of 30:0 and 0:30 mice. **e** Mean (±s.e.m.) spine head diameter in presumptive D1 and D2-MSNs from 30:0 and 0:30 mice ($n = 4$ animals per group; each data point is the mean head diameter of 6–12 neurons from each animal); $F_{(1, 12)} = 6.804$, $p = 0.0229$, main effect of *Training* in Two-way ANOVA; *$P = 0.0174$, post-hoc test. **f** Mean (±s.e.m.) spine head density in presumptive D1 and D2-MSNs from 30:0 and 0:30 mice ($n = 4$ animals per group; each data point is the mean head density of 6-12 neurons from each animal); $F_{(1, 12)} = 0.4$, $p = 0.5192$, main effect of *Training* in Two-way ANOVA. **g** Cumulative frequency (%) of spine head diameters (μm) in presumptive D1 and D2-MSNs from 30:0 and 0:30 mice; $F_{(3,2281)} = 15.34$, $p < 0.0001$, One-way ANOVA; ***$P < 0.0001$, GFP+ MSNs from 30:0 mice compared with GFP+ MSNs from 0:30 mice. **h** Graphical representation of virus delivery to the aDLS of D1-Cre and D2-Cre mice to express GCaMP6M in a Cre-dependent manner and collection of calcium events using miniscopes (left panels). Also represented is the identification and processing of regions of interest (ROIs) from calcium imaging data, and representative calcium traces from ROIs collected during the post-learning consolidation period (right panels). **i**, Mean (±s.e.m.) number of calcium events per minute identified in the aDLS of 10:20 and 30:0 D1-Cre mice during the post-magazine training period (126 ROIs from $n = 3$ 10:20 mice and 129 ROIs from $n = 3$ 30:0 mice) and during the post-acquisition period (129 ROIs from $n = 3$ 10:20 mice and 130 ROIs from $n = 3$ 30:0 mice); **$P_{(t = 2.933, \, df = 256)} = 0.0037$ compared with post-magazine period, unpaired two-tailed t-test in 30:0 mice. **j** Mean (±s.e.m.) number of calcium events per minute identified in the aDLS of 10:20 and 30:0 D2-Cre mice during the post-magazine training period (40 ROIs from $n = 3$ 10:20 mice and 54 ROIs from $n = 3$ 30:0 mice) and during the post-acquisition period (56 ROIs from $n = 3$ 10:20 mice and 54 ROIs from $n = 3$ 30:0 mice); **$P_{(t = 3.328, \, df = 94)} = 0.0012$ compared with post-magazine period, unpaired two-tailed $t$ test in 10:20 mice; ***$P_{(t = 5.080, \, df = 106)} < 0.001$ compared with post-magazine period, unpaired two-tailed $t$ test in 30:0 mice.

not 6 h later, dramatically enhanced performance of the new action in the retention test (Fig. 5f). Immediate but not delayed post-learning inhibition of D2-MSNs in aDLS markedly increased the number of response bouts (Fig. 5g), and modestly increased bout density (Fig. 5h), during the retention test. Cholinergic

interneurons in the striatum also express dopamine D2 receptors[38], raising the possibility that post-learning inhibition of cholinergic neurons in aDLS contributed to the increased consolidation observed in D2-Cre mice. To explore this possibility, we injected AAV8-hSyn-DIO-hM4Di-mCherry into

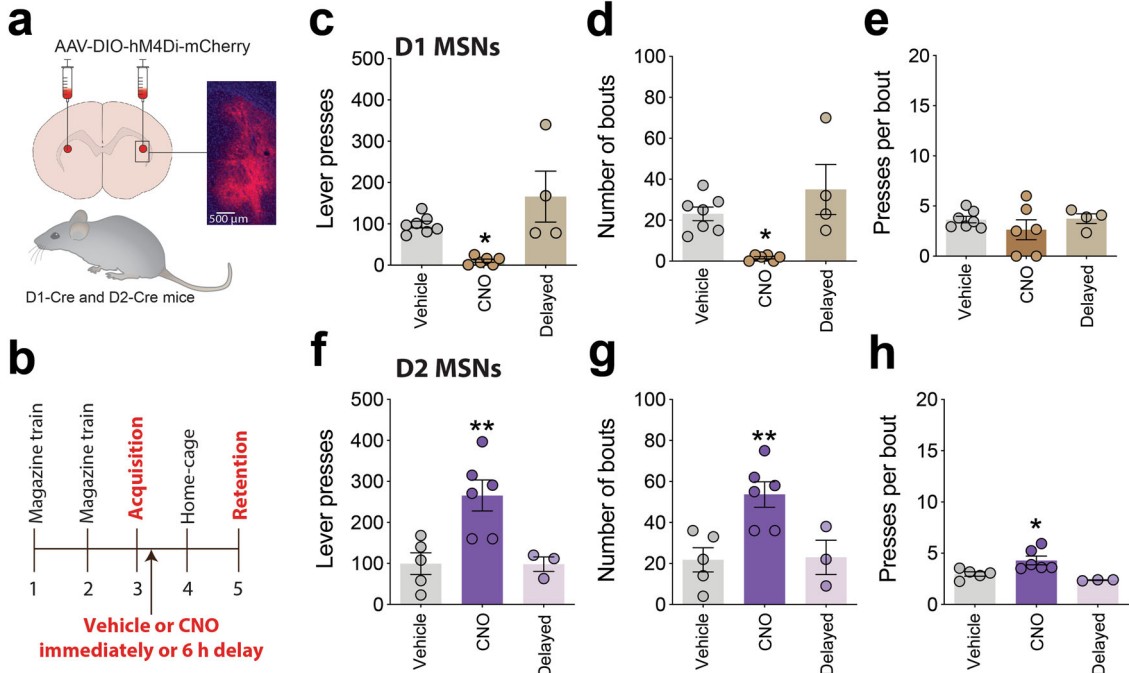

**Fig. 5 D1-MSNs in aDLS consolidate new learning and D2-MSNs oppose this process. a** Graphical representation of inhibitory Cre-dependent DIO-hM4Di-mCherry DREADD virus delivered to aDLS of D1-Cre and D2-Cre mice, and representative mCherry image from aDLS of virus-injected mouse. **b** Graphical representation of experimental design. **c** Mean (±s.e.m.) number of lever presses in D1-mice injected with vehicle ($n = 7$ mice), CNO soon after new learning ($n = 6$ mice), or CNO 6 h after new learning ($n = 4$ mice); $F_{(2, 14)} = 8.609$, $p = 0.0036$, One-way ANOVA; *$P = 0.0469$ compared with vehicle-treated mice, post-hoc test. **d** Mean (±s.e.m.) number of bouts of lever presses; $F_{(2, 14)} = 9.104$, $p = 0.0029$; *$P = 0.0226$ compared with vehicle-treated mice, post-hoc test. **e** Mean (±s.e.m.) number of lever presses per bout; $F_{(2, 14)} = 0.8103$, $p = 0.4645$. **f** Mean (±s.e.m.) number of lever presses in D2-mice injected with vehicle ($n = 5$ mice), CNO soon after new learning ($n = 6$ mice), or CNO 6 h after new learning ($n = 3$ mice); $F_{(2, 11)} = 8.944$, $p = 0.0049$, One-way ANOVA; **$P = 0.0081$ compared with vehicle-treated mice, post-hoc test. **g** Mean (±s.e.m.) number of bouts of lever presses; $F_{(2, 11)} = 8.112$, $p = 0.0068$, One-way ANOVA; *$P = 0.0099$ compared with vehicle-treated mice, post-hoc test. **h** Mean (±s.e.m.) number of lever presses per bout (±s.e.m.); $F_{(2, 11)} = 7.860$, $p = 0.0076$, One-way ANOVA; *$P = 0.0354$ compared with vehicle-treated mice, post-hoc test.

the aDLS of ChAT-IRES-Cre knock-in mice, which express Cre recombinase in cholinergic neurons, waited >3 weeks then trained the mice in the instrumental learning procedure (Supplementary Fig. 17). Neither immediate nor delayed (6 h) post-learning CNO injection altered the total number of lever-presses during the subsequent retention test, although there was a non-statistically significant trend for increased response bout frequency (Supplementary Fig. 17). Overall, these findings suggest that D1-MSNs in aDLS consolidate new instrumental actions whereas D2-MSNs and perhaps cholinergic interneurons oppose this process.

**Striatopallidal neurons in dorsolateral striatum regulate expression of habitual actions.** The aDLS plays a well-established role in regulating the expression of reward-independent habitual actions[15]. Therefore, we next explored whether the same region of the aDLS targeted in our experiments, which consolidates newly learned instrumental actions, also regulates the expression of habitual responses. To accomplish this, we first trained mice expressing hM4Di-mCherry in the aDLS (same mice as shown in Fig. 5) to lever-press for food rewards on a random interval (RI) schedule of reinforcement. Under this schedule, each lever-press delivered a food reward on average only once every 60 sec (RI60), which maintains high rates of lever-pressing yet delivers low numbers of rewards. This renders behavioral performance poorly correlated with reward delivery and thereby facilitates the emergence of goal-independent, habitual patterns of responding[39]. We found that vehicle-treated hM4Di-mCherry mice trained on the RI60 schedule responded in a habitual manner for food pellets, reflected by the fact that their lever-pressing behavior was unaffected by sensory-specific satiety-induced

devaluation of the pellets (Fig. 6b; see "Methods"). By contrast, CNO-treated hM4Di-mCherry mice reduced their responding only when food pellets were devalued but not when they were still valued (Fig. 6b). This suggests that aDLS inhibition reduced the expression of habitual actions and restored the ability of mice to respond in a flexible reinforcer-dependent manner, confirming previous reports[14]. Interestingly, the CNO-treated hM4Di-mCherry mice showed reduced frequency of response bouts when food was devalued (Fig. 6c), but no change in their number of magazine entries (Fig. 6d), density of response bouts or number of solitary lever-presses (Supplementary Fig. 18). This suggests that response bouts and reward retrieval behaviors, which were closely linked when mice learned to respond for food rewards under an FR1 schedule (see Supplementary Fig. 3), are uncoupled when mice respond in a habitual manner. These findings confirm that the same region of the aDLS that consolidates new instrumental actions also controls the expression of habitual actions.

Finally, we investigated whether the same cellular processes in aDLS that consolidate new instrumental actions also regulate the expression of habitual actions. We trained D1-Cre and D2-Cre mice expressing DIO-hM4Di-mCherry in the aDLS in the same RI60 schedule of reinforcement as described above. Chemogenetic inhibition of D1-MSNs in aDLS had no effects on any aspect of responding under the RI schedule when food pellets were valued or devalued (Fig. 6e, f). Similarly, chemogenetic inhibition of D2-MSNs had no effects on responding under the RI schedule when food pellets were still valued (Fig. 6g). However, when food rewards were devalued, D2-MSN inhibition reduced responding under the RI schedule (Fig. 6g), with the frequency of response

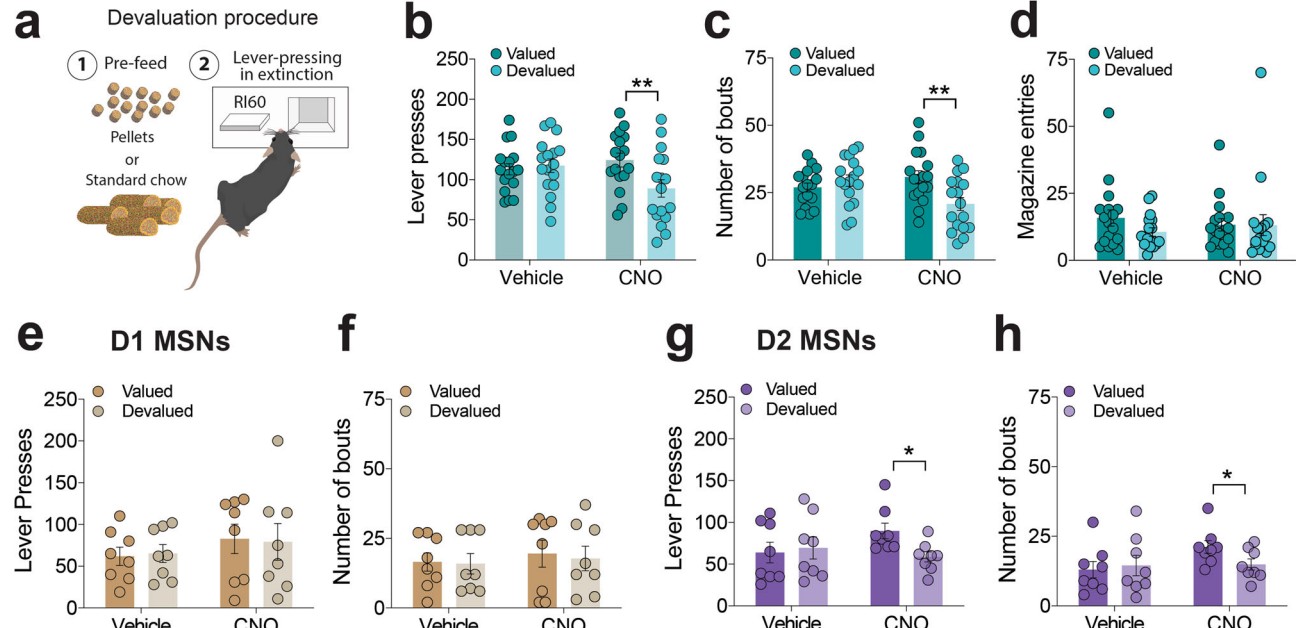

**Fig. 6 D2-MSNs in aDLS regulate habitual actions. a** Graphical representation of sensory-specific satiety-induced devaluation procedure used to assess the goal-directedness of lever-pressing behavior. Mice were pre-fed with chow (valued condition) or pellets (devalued condition) (1), then permitted to lever-press under extinction conditions (2). **b** Mean (±s.e.m.) number of lever presses under extinction conditions in mice expressing hM4Di in aDLS ($n = 22$) trained under an RI60 schedule after vehicle or CNO injection in valued and devalued conditions; $F_{(1,16)} = 13.84$, $p = 0.0019$, interaction between *Value* and *CNO* in Two-way repeated measures ANOVA; **$p = 0.0035$, post-hoc test. **c** Numbers of bouts of lever presses (±s.e.m.); $F_{(1,16)} = 17.56$, $p = 0.0007$, interaction between *Value* and *CNO* in Two-way repeated measures ANOVA; **$p = 0.0017$, post-hoc test. **d** Mean (±s.e.m.) number of magazine entries (±s.e.m.); $F_{(1,16)} = 1.807$, $p = 0.1977$, interaction effect in Two-way repeated measures ANOVA. **e** Mean (±s.e.m.) number of lever presses under extinction conditions in D1-Cre mice expressing hM4Di in aDLS ($n = 8$) trained under an RI60 schedule after vehicle or CNO injection in valued and devalued conditions; $F_{(1,7)} = 0.124$, $p = 0.7351$; interaction effect in Two-way repeated measures ANOVA. **f** Mean (±s.e.m.) number of bouts of lever presses; $F_{(1,7)} = 0.1076$, $p = 0.7525$. **g** Mean (±s.e.m.) number of lever presses under extinction conditions in D2-Cre mice expressing hM4Di in aDLS ($n = 8$) trained under an RI60 schedule after vehicle or CNO injection in valued and devalued conditions; $F_{(1,7)} = 0.6359$, *$p = 0.0397$; interaction effect in Two-way repeated measures ANOVA. **h** Mean (±s.e.m.) number of bouts of lever presses; $F_{(1,7)} = 6.055$, *$p = 0.434$; interaction effect in Two-way repeated measures ANOVA.

bouts (Fig. 6h) but not their density (Supplementary Fig. 19) being the feature of behavior that was selectively decreased. These findings suggest that D2 but not D1-MSNs in the aDLS regulate the expression of habitual actions. This is consistent with recent observations suggesting that D2-MSNs in aDLS undergo synaptic remodeling as habitual patterns of responding emerge[40] and that D2 receptor antagonists attenuate the expression of habit-like actions[41]. Together, these data suggest that D1-MSNs in aDLS regulate the consolidation of newly learned instrumental actions whereas D2-MSNs in the aDLS oppose this consolidation process and instead promote the expression of previously learned habitual actions.

## Discussion

In comparison to our considerable understanding of the molecular, cellular and circuit mechanisms of Pavlovian and other forms of learned associations[42–45], much less is known about how new instrumental associations are encoded in the brain. Here, we show that the aDLS region of the striatum plays a critical role in consolidating new instrumental actions. The aDLS is known to regulate the acquisition and expression of stimulus-response associations, in which conditioned stimuli in the environment come to elicit a behavioral response independent of any representation of the reward that originally reinforced that response[15]. Such value-free stimulus-response learning contributes to the development and persistence of habitual actions[46]. However, new instrumental learning often coincides with new sensorimotor

learning, suggesting that some of the same brain systems may participate in both types of behavioral plasticity[18]. Our data support this hypothesis by establishing an important role for the aDLS in regulating new instrumental conditioning. Our data also reveal a striking partitioning of cellular function in the aDLS in which D1-MSNs consolidate newly acquired instrumental actions and D2-MSNs oppose this consolidation process and instead promote the expression of habitual actions. Finally, our data suggest that the aDLS acts in concert with the NAc during new instrumental conditioning to encode information relevant to reinforcer magnitude and effort requirements, respectively.

In vivo cellular recordings have established that neurons in the aDLS are active during the earliest stages of new instrumental learning in rodents[16,18,32,47,48]. The putamen region of the human brain, considered equivalent to the rodent DLS, is similarly active during new instrumental learning[10,49]. Based on these findings, it was hypothesized that the DLS may participate in learning new instrumental responses in addition to its well-established role in stimulus-response learning[18]. Accordingly, we found that blockade of new protein synthesis in the aDLS of rats soon after the successful acquisition of a new instrumental action, but not 6 h later, reduced the subsequent expression of the response when retention of the new memory was tested 2 days later. Similarly, chemogenetic inactivation of the aDLS soon after instrumental learning, or inhibition of only those ensembles of aDLS neurons active soon after instrumental learning, impeded the ability of mice to consolidate the new instrumental action. The frequency of response bouts was the feature of behavior that

was most reliably impacted in rats and mice by post-learning aDLS manipulations. Propensity to engage in bouts of responding is thought to be influenced by the value of the reinforcer delivered by that same instrumental response[24–27]. Consistent with this link, we found that the propensity to engage in response bouts during a retention test was related to the value of the reinforcer available during the training session (sucrose versus chow pellets). Hence, the aDLS likely encodes information related to the perceived value of a new response during instrumental conditioning. This is surprising in light of the well-established role of the aDLS in regulating habitual actions that are executed in a value-independent manner[15]. As described in more detail below, this apparent discrepancy likely reflects the dissociable contributions of D1 and D2-MSNs in the aDLS to influence value-dependent and -independent behaviors, respectively.

Most previous studies investigating striatal mechanisms of instrumental conditioning have focused on the role of the NAc in this process[12,32,50–52]. Infusion of the protein synthesis blocker anisomycin into the NAc of rats blocked the acquisition of a new lever-press response that delivered food rewards when the infusions occurred immediately but not >2 h after each daily training session[7]. Similarly, pharmacological blockade of NMDA or AMPA glutamate receptors[53–56], D1 dopamine receptors[54,55] or muscarinic acetylcholine receptors[57] in the NAc prevented the acquisition of a new instrumental response in rats, reflected by low levels of lever-pressing for food rewards across training sessions compared with vehicle-treated rats. Excitotoxic lesion of the NAc core also disrupted the acquisition of a lever-press response in rats[8]. However, recent observations support a more nuanced reinterpretation of these data. For example, anisomycin infused into the NAc can trigger aversion to food items consumed soon afterwards[11], suggesting that this manipulation may not block consolidation per se but instead decrease the perceived value of recently consumed food rewards. This, in turn, would be expected to reduce the willingness of animals to engage in a new instrumental action that delivered a devalued food reward independent of any abnormalities in the consolidation process. Consistent with this interpretation, intra-NAc anisomycin infusion had no effects on consolidation of a new instrumental response when its inhibitory effects on food reinforcer valuation were prevented using a sucrose protection procedure[11]. Furthermore, glutamatergic, dopaminergic and muscarinic receptor antagonists disrupted the execution of a new instrumental action only when infused into the NAc before each daily training sessions but had no effects when infused after training[55]. This suggests that neurotransmission in the NAc regulates the performance of behaviors necessary to learn new instrumental actions but not the post-learning consolidation processes. Similarly, lesions of the NAc disrupted new instrumental learning only when a delay was imposed between execution of the new action and delivery of the reward[8], with lesioned animals learning at normal rates when food rewards were delivered without delay[8]. This is consistent with proposals that the NAc, and dopaminergic transmission in this site, regulates effort-based decision-making[58]. According to these proposals, the NAc regulates willingness to engage in actions depending on the costs associated with obtaining associated rewards, such as the amount of effort that must be invested or the length of a delay that must be endured after executing an action but before reward delivery occurs[8,58–60]. Consistent with these observations, we found that disrupting protein synthesis in the NAc soon after new instrumental learning did not block the subsequent expression of the new response, although there was a non-statistically significant trend for decreased total lever-pressing during the retention test. Instead, post-learning NAc infusions of anisomycin reduced the vigor of executing the new response, reflected by decreased density of response bouts. As

noted above, bout density is thought to depend on the amount of effort required to obtain a reward, with increases in bout density occurring when greater effort is required[61]. Consistent with this interpretation, we found that increasing the amount of effort necessary to earn a food reward during the training session (FR2 instead of FR1 schedule) increased bout density during the subsequent retention test. Hence, our findings support an important role for the NAc in encoding performance-relevant information during instrumental conditioning. This contrasts with the effects of post-learning infusions of anisomycin into the aDLS, which specifically decreased the frequency of response bouts during the retention test, suggesting that the aDLS encodes information relevant to reinforcer value during instrumental conditioning. As the frequency and density of response bouts are dissociable aspects of the same instrumental response, this implies that the aDLS and NAc must act in a coordinated fashion during instrumental conditioning to encode discrete but closely related aspects of the same action sequence. It is unclear how such cooperativity is organized, but one possibility is that the NAc is functionally linked to the aDLS and other dorsal regions of the striatum through ascending or so-called "spiraling" reciprocal connections between MSNs and midbrain dopamine neurons[62,63]. Whatever the underlying mechanisms, our findings support key roles for the aDLS and NAc in consolidating new instrumental actions and suggest that close interactions between these striatal regions is necessary for successful consolidation to occur.

Using in vivo calcium imaging, we detected increased activity of D1-MSNs in the aDLS during the post-learning phase when a new instrumental response was consolidated, whereas D2-MSN activity was decreased during this same period. Chemogenetic inhibition of D1-MSNs in aDLS during this period almost completely ablated the ability of mice to encode the new action, whereas post-learning inhibition of D2-MSNs dramatically strengthened this process. This suggests that D1 and D2-MSNs in the aDLS exert functionally antagonistic effects on the consolidation of new instrumental actions. Inhibition of D2-MSNs in the aDLS also decreased the expression of (value-independent) habitual responses in mice, whereas inhibition of D1-MSNs had no effects on habitual responding. These findings suggest that D1-MSNs in aDLS specialize in encoding new instrumental actions whereas D2-MSNs in aDLS specialize in habitual actions, consistent with recent reports[40,41]. This portioning of function between D1 and D2-MSNs provides a parsimonious explanation for how the aDLS can regulate both value-dependent and -independent behaviors. Considering that inhibition of D2-MSNs in the aDLS dramatically strengthened the consolidation of new instrumental learning while also blocking habitual responding, it is likely that this cell population is the ultimate arbiters of whether new instrumental actions are encoded or previously learned habitual actions are expressed. Indeed, D2-MSNs are known to send provide short-range inhibitory inputs to neighboring D1-MSNs[64,65], providing a potential mechanism through which D2-MSNs can regulate D1-MSN activity to control new instrumental learning.

Finally, it is noteworthy that both D1 and D2-MSNs in the DLS regulated the propensity to engage in bouts of responding, with each cell type assuming control over bout frequency in different behavioral contexts. D1-MSNs regulated bout frequency when animals first learned a new instrumental response that delivered an unexpected reward, whereas D2-MSNs regulated bout frequency when animals expressed a previously learned action in a habitual manner. As noted above, the frequency of response bouts appears to be related to the relative value of the action[24–27]. If so, this would suggest that MSNs in the aDLS serve to attribute value to actions when there is some ambiguity about the precise

relationship between the action and its outcome. More specifically, D1-MSNs may attribute value to novel action sequences that unexpectedly deliver rewards to promote new instrumental learning, whereas D2-MSNs may attribute value to previously learned actions expressed autonomously because they deliver rewards in an unpredictable manner. Consistent with this possibility, neural activity related to reward, action, and choice all converge in the DLS when the consequences of an action are ambiguous at the time of its execution, most prominently when there is a delay between action and reinforcer delivery[66].

In summary, our data support an important role for D1-MSNs in the aDLS in consolidating newly learned instrumental actions and suggest that D2-MSNs in aDLS oppose this new learning and instead promote the expression of previously learned habitual actions. These findings have important implications for understanding how actions are acquired, stored, and expressed by the striatum. Those suffering from disorders associated with abnormalities in striatal function, including Parkinson's disease, schizophrenia, and drug addiction, often show altered instrumental and/or habit learning. Better understanding of the role of D1 and D2-MSNs in the aDLS may reveal new insights into the pathophysiology of these disorders.

## Methods

**Animals**. For all rat experiments, we used male Long Evans rats weighing 275–300 g, purchased from Charles River Laboratories and housed 2 per cage. For mouse experiments, male C57Bl6/J mice, or transgenic strains congenic with a C57 background, were used. Animals were housed in groups 2 per cage for rats or 2-4 per cage for mice in an environmentally controlled vivarium on a 12-h:12-h reversed light:dark cycle, with food and water available *ad libitum* until behavioral training commenced. During behavioral training, rats and mice were food restricted to 85–93% of their free-feeding body weight, while water was provided *ad libitum*. All rats weighted at least 300 g, and mice were aged at least 2-3 months, at the start of experiments. Transgenic mice used were: D1-Cre (MMRRC Stock No. 37156-JAX, gift from Dr. Eric Nestler); D2-Cre (MMRRC Stock No. 017263-UCD, gifts from Drs. Eric Nestler and Scott Russo); ChAT-Cre (Stock No 031661-JAX); Fos-Cre$^{ERT2}$ mice (Stock No. 021882, Jackson Laboratories, Bar Harbor, ME), Fos$^{2A\text{-iCreER}}$ (Stock No 030323; Jackson Laboratories, Bar Harbor, ME). All animals were group housed in a reverse 12 h light cycle (lights on 7 pm). Around 21 days of age, mouse pups were weaned and ~1 mm of tail was clipped for genetic analysis. DNA was extracted with a tissue DNA extracted (Biomiga, Inc. San Diego, CA). Primers for Drd1-Cre were: 5′-GAACCTGATGGACATGTTCAGG-3′ and 5′-CGGCAAACGGACAGAAGCATT-3′. Drd2-Cre: 5′-AGTGCGTTCGAACGC TAGAGCCTGT-3′ and 5′-CGGCAAACGGACAGAAGCATT-3′. Chat-Cre: 5′-GCAAAGAGACCTCATCTGTGGA-3′, 5′-GATAGGGGAGCAGCAACAAG-3′, 5′-TTCACTGCATTCTAGTTGTGGT-3′. Fos-Cre$^{ERT2}$: 5′-CACCAGTGTCTA CCCCTGGA-3′ and 5′-CGGCTACACAAAGCCAAACT-3′ (wild-type reverse), or 5′ CGCGCCTGAAGATATAGAAGA-3′ (mutant). Fos$^{2A\text{-iCreER}}$: 5′-GTCCGG TTCCTTCTATGCAG-3′, 5′-GAACCTTCGAGGGAAGACG-3′, 5′-CCTTGCAA AAGTATTACATCACG-3′. See Supplemental Table 2 for all primers used. Samples underwent DNA amplification according to the standard PCR protocol from Jackson Labs, then run on a 2% agarose gel containing 0.5 μg/mL ethidium bromide. Gels were imaged under ultraviolet light on a Gel Doc XR+ (Bio-Rad, Bio-Rad, Hercules, CA). Transgenic bands were at 340 bp for the Drd1-Cre mice, 700 bp for Drd2-Cre mice, 250 bp for Chat-Cre mice, 300 bp for Fos-Cre$^{ERT2}$ mice, and 230 bp for Fos$^{2A\text{-iCreER}}$ mice. All animal husbandry and behavioral procedures were conducted in strict accordance with the NIH Guide for the Care and Use of Laboratory Animals and procedures were approved by the Institutional Animal Care and Use Committees of The Scripps Research Institute or the Icahn School of Medicine at Mount Sinai.

**Drugs**. Anisomycin (Sigma) was dissolved in an equimolar concentration of HCl, adjusted to pH 7.2, brought to a concentration of 125 μg/μl in PBS, and microinjected intracranially in rats at a volume of 0.5 μL over 1 min, resulting in an effective dose of 62.5 μg/side. A mixture of the GABA$_A$ receptor agonist muscimol (mus) and GABA$_B$ receptor agonists baclofen (bac) was dissolved in PBS at the doses of 0.3 nmol and 0.03 nmol per 0.5 μL and microinjected intracranially in rats at a volume of 0.5 μL per side over 1 min. Clozapine-N-oxide (CNO, Enzo Life Sciences, Farmingdale, NY) was diluted in 0.9% saline for intraperitoneal injection at doses or 3 or 1 mg kg$^{-1}$, at a volume of 1 ml per 100 g of body weight. 4-Hydroxytamoxifen (Sigma-Aldrich) was prepared in corn oil and injected at a dose of 50 mg kg$^{-1}$, at a volume of 1 ml per 100 g of body weight.

**Virus vectors**. For all DREADD experiments commercially available adeno-associated virus (AAV) particles were purchased from Addgene. We used non-Cre-dependent hM4Di (AAV8-hSyn-HA-hM4D(Gi)-IRES-mCitrine; Addgene Cat# 50464) for non-cell type-specific inactivation of the aDLS in mice. To inhibit activity of D1 or D2-MSNs in aDLS, we used AAV8-hSyn-DIO-hM4D(Gi)-mCherry (Addgene Cat# 44362). To manipulate neuronal ensembles in FosCre$^{ERT2}$ mice, we used AAV8-hSyn-DIO-HA-hM4Di-IRES-mCitrine (Addgene cat# 50455) and AAV8-hSyn-DIO-hM3D(Gq)-mCherry (Addgene Cat#44361). For calcium imaging experiments we used AAV-DJ-EF1a-DIO-GCaMP6m from Stanford Vector Core (Palo Alto, CA). All viruses were distributed into 5 μL aliquots, kept at −80 °C, and thawed immediately before injection.

**Intracranial implantation and microinjections**. Mice and rats were anesthetized using an isoflurane (3–5% induction, 1–3% maintenance) and positioned in a stereotaxic frame (Kopf Instruments, Tujunga, CA). Prior to beginning stereotaxic measurements, flatness of the skull was ensured by observing identical dorsal/ ventral coordinates at each bregma and lambda. To test the effect of intracranial infusion into domains of the striatum of anisomycin or other agents on consolidation or expression of a new instrumental response by rats, we implanted bilateral 22-gauge, stainless-steel guide cannula (Plastics One, Wallingford, CT) 1.0 mm above the anterior dorsolateral striatum (coordinates from bregma: anterior-posterior: +1.5 mm, medial-lateral: ±3.5 mm, dorsal-ventral from skull: −3.8 mm), 1.0 mm above the posterior dorsolateral striatum (coordinates from bregma: anterior-posterior: +0.5 mm, medial-lateral: ±3.5 mm, dorsal-ventral from skull: −3.8 mm), 1.0 mm above the posterior dorsomedial striatum (coordinates from bregma: anterior-posterior: −0.26 mm, medial-lateral: ±1.75 mm, dorsal-ventral from skull: −3.8 mm), or 2.5 mm above the core of the nucleus accumbens (coordinates from bregma: anterior-posterior: +1.5 mm, medial-lateral: ±1.75 mm, dorsal-ventral from skull: −4.7 mm). Standard stereotaxic techniques were used, and the cannula was secured to the skull using bone screws and dental cement. Twenty-nine-gauge obturators flush with the end of the guide cannula (Plastics One, Roanoke, VA) were inserted in the guide cannula. After surgery, rats were allowed a recovery period of at least 7 days before behavioral testing. Before the start of experimental infusions, the rats were habituated to the infusion procedure with a surrogate infusion, which consisted of the removal and replacement of the obturator during gentle restraint within a time course identical to that of drug infusion. During infusions, the rats were gently restrained while the obturators were removed and a 29-gauge bilateral injector, which protruded 1.0 mm (anterior dorsolateral striatum, posterior dorsolateral striatum, posterior dorsomedial striatum) or 2.5 mm (core of the nucleus accumbens) below the end of the guide cannula, was inserted, and 0.5 μl solution infused over a 1 min period into targeted site. The injector was left in place for 2 min to allow the drug to diffuse in the local vicinity of the injector tip. The injector was then carefully removed, and the obturator replaced. All infusions were delivered in the behavioral testing room. To test the effects of chemogenetic inhibition of the aDLS on consolidation of new instrumental learning in mice, synthetic receptors (Cre-dependent and non-Cre-dependent M4 DREADDs) were delivered via AAV vectors to the aDLS of wild-type of Cre-expressing mice. Bilateral injections (0.3 μL per side at a flow rate of 0.1 μL per min) were made at the following coordinates from bregma: anterior-posterior: +1.3 mm, medial-lateral: ±2.0 mm, dorsal-ventral: −3.1 mm. The injector needle remained in place for 5 min after injection. Following injection, mice were allowed to recover for at least 2 weeks before experimentation.

**Brain perfusion and fixation**. Mice and rats were deeply anesthetized with an isoflurane and perfused through the ascending aorta with 0.1 M PBS, followed by 4% paraformaldehyde (PFA) in 0.1 M PBS (pH 7.4). The brains were removed and post-fixed in paraformaldehyde. Prior to being sectioned, the brains were transferred to 20% sucrose in 0.2 M phosphate buffer and left overnight. To confirm intracranial injection sites in rats, coronal sections were cut at 60 μm on a freezing microtome and stained with Cresyl Violet. Cannula locations were mapped onto standardized sections of the rat brain (Paxinos and Watson, 1998), and investigated for any signs of tissue damage. To confirm virus injection sites in mice, post-fixed brains were transferred to 20% sucrose solution in 0.2 M phosphate buffer and left overnight. Coronal sections were cut at 40 μm on a freezing microtome and imaged for GFP, mCitrine, or mCherry expression.

**Whole-brain c-Fos mapping in mice**. Wild-type male C57BL6/J mice were food restricted to 88–93% of free-feeding body weight and were trained to lever-press for food rewards under a FR1 schedule of reinforcement during a single session as described below. 60 min after they earned the final food pellet in the training session mice were perfused with 4% paraformaldehyde and their brains post-fixed overnight at 4 ºC. Tissue clearing and c-Fos staining were performed according to the detailed protocol available at http://idisco.info. Briefly, brains were dehydrated, bleached to reduce background autofluorescence, rehydrated, then incubated in rabbit anti-c-Fos antibody (1:1000, Synaptic Systems Cat# 226–003) for 7 days, followed by washing and incubation in AlexaFluor 790 nm conjugated anti-rabbit secondary antibody (1:1000, Invitrogen Cat# A11374) for 7 days. Following additional washing, brains were dehydrated and cleared in methanol and dichloromethane, then refractive index matching occurred in dibenzyl ether. Brains

were imaged at 1.3× magnification, 0.1 numerical aperture, in the horizontal orientation on a LaVision Ultramicroscope II light-sheet microscope with a near-isotropic xyz resolution of 5 μm × 5 μm × 4.5 μm. Images were acquired with a 488 nm laser for an autofluorescence reference channel, and a 785 nm laser for acquisition of secondary antibody fluorescence. The ClearMap Python package (www.github.com/christophkirst/clearmap) was used for cell detection and registration of cell coordinates onto the Allen Brain Atlas. Data analysis occurred on a Dell Precision T7810 Workstation running Ubuntu 18.04LTS, and followed steps outlined in ClearMap documentation. Briefly, data were downsampled and the Elastix toolbox (http://elastix.isi.uu.nl/) was used to perform automated 3D affine and B-spline transformation to register the autofluorescence signal channel to the 25 μm resolution Allen Brain Atlas reference, and to correct for any motion between imaging of the autofluorescence and signal channel images. Prior to cell detection, background subtraction occurred using a morphological opening of 8 × 8 pixels, and Difference of Gaussian feature enhancement filter with a Gaussian kernel size of 4 × 4 × 4 pixels was applied. Cells were detected with a peak intensity threshold of 125. Cell objects were painted via a watershed until reaching this threshold, and only cells with sizes between 8 and 200 continuous pixels were included. The Transformix module of the Elastix toolbox was then used to apply transformation vectors from the registration step to cellular coordinates, and cell counts for each region were calculated. In order to parse the caudoputamen into subregions (aDLS, pDMS, etc.), the ClearMap automated isolation function was used to create and export a NumPy array containing coordinates of cells in the caudoputamen. Custom Python scripts were then used to bin these coordinates along the anterior/posterior, and then medial/lateral axes. Custom scripts, as well as example parameter and process scripts containing all detailed cell detection parameters and thresholds, are available at www.github.com/kennylabsinai/SmithJonkmanNatComm2021.

**c-Fos expression after instrumental learning in rats**. Rats were food restricted to 88–93% of free-feeding body weight and trained to lever-press for food rewards under a FR1 schedule of reinforcement during a single session as described below. 60 min after they earned the final food pellet in the training session rats were perfused with 4% paraformaldehyde and their brains post-fixed overnight at 4 °C. Brains were incubated in 20% sucrose until they sank completely in solution (48–72 h). A freezing cryostat was used to collect 40 μm sections through the aDLS, pDLS, aDMS, pDMS, and NAc. Free-floating sections were permeabilized in 0.1% TritonX-100 in PBS and then blocked in 5% normal donkey serum (NDS). Immunofluorescence labeling was performed using anti c-Fos antibody at 1:1000 dilution (Synaptic Systems, Gottingen, Germany; Cat# 226003) for 16 h. Sections were washed three times for 10 min in PBS containing 0.2% Tween-20 (PBST). Donkey-Anti-Rabbit Alexafluor594 secondary antibody was used at 1:1000 dilution (Jackson Immunoresearch, Bar Harbor, ME; Cat# 711-585-152). Sections were washed in PBST x3 times for 10 min each, mounted on Fisherbrand Superfrost slides, then imaged with a Zeiss Axiophot 2 epifluorescene microscope. Cells were counted using ImageJ software.

**Quantification of dendritic spine morphology**. D1-Cre mice were injected into the aDLS with AAV-DIO-GFP to label D1-MSNs and three weeks later were trained to acquire a new lever-press response as described below. 15 min after they earned the final food pellet in the acquisition session the mice were deeply anesthetized with isoflurane, then transcardially perfused with 20 mL cold 0.1 M phosphate buffer (PB) followed by 20 mL 1.5% PFA in PB. Brains were removed and post-fixed in the same fixative for 30 min, then coronally sectioned at 200 μm in PBS on a vibratome. Tungsten particles (1.3 μm diameter, Bio-Rad) were coated with the lipophilic carbocyanine dye DiI (Invitrogen). DiI-coated particles were delivered diolistically into the tissue at 80 PSI using a Helios Gene Gun system (Bio-Rad) fitted with a polycarbonate filter with a 3.0 μm pore size (BD Biosciences). DiI was allowed to diffuse along neuron axons and dendrites in PB for 24 h at 4 °C, and then fixed again in a 4% PFA for 1 h at room temperature. After a brief PB wash, tissue was mounted onto slides in aqueous medium Prolong Gold (Invitrogen). For spine morphology analysis, images of DiI-labeled sections were taken on a confocal microscope (Zeiss) using a Helium/Neon 543 nm laser line. Optimal sampling frequency was calculated using the Nyquist-Shannon sampling theorem. Images of dendrites were taken through a 63x oil immersion objective (Plan-Apochromat, Zeiss; NA = 1.4, WD = 90 μm) with pixel size 0.07 μm in the XY-plane and 0.10 μm intervals along the Z-axis. Images were deconvolved via Autoquant x3 prior to analysis (Media Cybernetics, Bethesda, MD), and spine head morphology was quantified via NeuronStudio (http://research.mssm.edu/cnic/tools-ns.html). Only spines on dendrites beginning >75 μm and ending <200 μm distal to the soma and after the first branch point were quantified on cells localized to the aDLS. The length of quantified segments was 45–55 μm. One segment from each neuron was quantified, and the minimum spine head diameter was set at 0.15 μm. Between 6 and 12 neurons were imaged in each animal.

**In vivo calcium imaging of D1 and D2-MSNs**. Miniature fluorescent microscopes and data acquisition electronics were built using parts lists, schematics, and instructions available at miniscope.org[37]. Data were collected at 15 frames per second, and LED power was scaled between 10 and 25% for optimal signal-to-

noise ratio. D1-Cre and D2-Cre mice were single housed for these experiments in order to prevent cage-mates from interfering with lens implants. Mice underwent two surgical procedures. First, they were received intra-aDLS injection of AAV-DJ-EF1a-DIO-GCaMP6m. One week later, ProView gradient-index (GRIN) lens that was 1 mm in diameter and 4 mm length (Part number 1050-002202, Inscopix, Palo Alto, California) was implanted immediately above the aDLS. During this procedure, a 1 mm craniotomy was created above the injection site, and the cortex directly below the craniotomy was aspirated with a 27-gauge blunt needle attached to a vacuum. Artificial cerebrospinal fluid was continuously applied to prevent drying of exposed tissue. In the case of prolonged bleeding, a blunted 30-gauge needle was used to remove blood with minimal additional tissue loss. Cortical tissue was aspirated until horizontal striations of the corpus callosum were clearly visible. A set screw was then implanted into the contralateral skull and the GRIN lens was slowly lowered and fixed in place using cyanoacrylate dental cement. Kwik-Sil low toxicity silicone adhesive (World Precision Instruments) was applied to the top of the GRIN lens to keep it clean. Mice were injected with the glucocorticosteroid dexamethasone (0.2 mg kg$^{-1}$, SC) at the time of surgery and daily for three days following surgery to minimize inflammation caused by the lens. Two weeks later, a metal baseplate with a miniscope attached was positioned over the GRIN lens until optimal focal plane was observed. The baseplate was then fixed in place using cyanoacrylate dental cement, and the miniscope was removed. A magnetic plastic cover was placed over the baseplate at all times except for during imaging. After recovery from surgery, and beginning 7 days prior to behavioral experiments, mice were habituated to miniscope attachment for 20 min daily, after which time no overt differences in movement or behavior resulting from head-mounting of miniscope were detectable. After habituation to the miniscope, mice were placed into the operant chamber and magazine training occurred, as described below. Immediately after the first magazine training session, miniscopes were head-mounted onto mice in their home cage, and calcium transients recorded for 15 min. Immediately following the acquisition session, calcium transients were again imaged in the home cage for 15 min during the period when consolidation of the lever-press response occurs.

**Single-session instrumental learning in rats and mice**. Experiments were carried out in sound-attenuated operant conditioning chambers (Med Associates, St. Albans, VT). The metal grid flooring was cleaned with a paper towel and 0.1% Micro 90 cleaning agent, and fresh bedding was placed in the tray underneath the grid, before and after every session. Rats and mice were mildly food restricted to 85–90% of their free-feeding body weight. They were placed into operant conditioning chambers with the house-light turned off, the levers retracted, and underwent magazine training sessions on consecutive days. During magazine training, 30 pellets (45 mg for rats, 20 mg for mice; TestDiet, Richmond, IN) were delivered noncontingently into the food magazine according to a variable time 60 s schedule of reinforcement. The next day, mice and rats were placed into operant conditioning chambers with the house-light off, and 1 min later the left lever was extended into the chamber and animals permitted to respond on the lever to earn food pellets, delivered into the food magazine, according to a fixed ratio 1 (FR1) schedule of reinforcement with a 1 s time-out between each pellet earned. After criterion numbers of pellets were earned (50 pellets for rats, 30 pellets for mice) the acquisition session was ended, and animals returned to their home-cage. Only animals that earned the criterion number of pellets within 90 min (rats) or 120 min (mice) advanced to the retention test (~90% of all animals). 48 h later they were placed into the operant conditioning chamber, with the house-light off, and 1 min later the left lever was extended into the chamber. Responding on the lever was recorded for 60 min but had no programmed consequence (i.e., extinction conditions). Consumption of all noncontingently delivered or earned pellets was visually confirmed for each animal after every session.

**Calculation of response bouts**. During the retention test in rats or mice, probability of a lever-press response in a given time interval range in 5 s epochs was calculated. During the first 0–5 s incremental epoch after a lever-press, the total responses that occurred after the preceding response was divided by the total intervals of at least 0 s. For the 5–10 s epoch, total responses within 5–10 s after a preceding response was divided by all inter response intervals of at least >5 s, and so on. Based on this calculation, the highest probability that a second lever-press response would occur after a preceding response was maximal in the 0–5 s epoch (see Supplementary Fig. 2). Therefore, about was defined as any two or lever-press responses that occurred within 5 s.

**Random interval 60 (RI60) schedule of reinforcement**. Mice that had learned to lever-press for food rewards under a FR1 schedule until they achieved stable levers of responding across sessions (>25 rewards per 60 min daily session) were trained on the RI60 schedule of reinforcement during 60 min sessions. At the beginning of each RI60 session, the lever was extended in an inactive state. Each second there was a 1 in 60 chance that the lever would become active. When active, the next lever press response resulted in reinforcer delivery and the lever returned to an inactive state. This resulted in an average time between reinforcer delivery of 60 s.

**Sensory-specific satiety-induced devaluation procedure**. Mice that responded for food pellets under a RI60 schedule of reinforcement were allocated to vehicle and CNO groups, counter-balanced based on treatment history, cage-mate (such that at least one mouse from each cage was in each condition), and the number of rewards earned during RI60 training. On the test day, mice were permitted 75 min access to standard laboratory chow (valued) or the same food pellet rewards earned during operant conditioning sessions (devalued), in their home cage. They were then injected with vehicle or CNO according to the experimental design (see below and Main text) and placed back into their home-cage. 15 min later they were placed into the operant conditioning chamber and the test session initiated. During the test session, mice lever-pressed for food rewards under extinction conditions under the or RI60 schedule (i.e., their responding had no scheduled consequences) for 15 min and their lever press responses and magazine entries were recorded. Each animal was tested twice, once in the valued and once in the devalued condition (in an order counterbalanced by RI responding), but both times under the same treatment condition (vehicle or CNO injection).

**Blockade of protein synthesis and sucrose protection procedure**. Bilaterally cannulated rats underwent new instrumental conditioning as described above. With some slight modifications. After the second magazine training session, rats were placed individually into a plexiglass cage with sawdust bedding and a water drinking bottle for 15 min in order to habituate them to a new environment in which they would receive sucrose or chow pellets. Immediately after the acquisition session, rats were again placed individually into a plexiglass cage with sawdust bedding and a drinking bottle containing a sucrose solution (20% w/v), to which they were naïve, for 15 min and their sucrose consumption was recorded by weighing the sucrose-containing drinking bottle before and after the session. Rats then received intra-striatal injection of PBS vehicle or anisomycin solution immediately after consuming the sucrose solution, or intra-striatal injection of anisomycin 6 h later. The assignment of rats to these treatment groups was counterbalanced for time required to earn 50 pellets during the acquisition session. After intra-striatal infusion, rats were returned to their home-cage and left undisturbed. 48 h later, rats were returned to the operant conditioning chamber for the retention test and immediately afterward were again placed individually into a plexiglass cage with sawdust bedding and a drinking bottle containing a 20% sucrose solution for 15 min. Sucrose consumption was again recorded by weighing the bottle before and after the free consumption session. In this manner we could determine whether intra-striatal anisomycin infusion caused aversion to of the sucrose solution. The next day, rats were placed in the operant conditioning chamber with the house-light off, and 80 food pellets were placed in the magazine. After 5 min, the rats were removed from the chamber, and the number of pellets consumed was recorded for each rat. In this manner we could determine whether intra-striatal anisomycin infusion caused any avoidance of the food pellets.

**Chemogenetic modulation of the aDLS**. To chemogenetically inhibit the aDLS in wildtype mice, AAV8-hSyn-hM4D(Gi)-mCitrine or a control AAV8-hSyn-GFP virus was stereotaxically injected into the aDLS and mice permitted at least 2 weeks for recovery before behavioral training commenced. To chemogenetically inhibit D1 or D2-MSNs, transgenic D1-Cre and D2-Cre mice, respectively, were stereotaxically injected into the aDLS with AAV8-hSyn-DIO-hM4D(Gi)-mCherry and permitted at least 2 weeks for recovery before behavioral training commenced. Following the acquisition session, mice were assigned to "vehicle", CNO, or delayed CNO groups, counterbalanced by time required to reach acquisition criteria. For devaluation testing, mice were injected with vehicle (PBS or saline) or CNO (1 or 3 mg kg$^{-1}$ dissolved in PBS), counterbalanced by responding during FR or RI60 training sessions, and testing commenced 15 min later. After injection and treatment sessions, mice were undisturbed for 48 h to allow for drug washout.

**Chemogenetic modulation of TRAPed neurons in the aDLS**

*Chemogenetic inhibition experiment.* Fos$^{2A-iCreER}$ (i.e., FosTRAP2) heterozygous mice were injected with AAV8-hSyn-DIO-HA-hM4D(Gi)-IRES-mCitrine (M4-DREADD; $n = 23$) into the aDLS and permitted 10 days to recover. These mice then underwent 2 consecutive days of magazine training (60 min sessions). Following the second magazine training session, mice received either vehicle (corn oil; $n = 18$) or 4-OHT (50 mg kg$^{-1}$, IP; $n = 5$; Sigma-Aldrich Cat# H6278) injection in order to TRAP a neuronal ensemble associated with non-contingent reward delivery. The next day mice were permitted to lever-press under a FR1 schedule until they earned 30 food pellets. Immediately following this acquisition session, mice were injected with either vehicle ($n = 6$) or 4-OHT ($n = 7$; all mice that received 4-OHT following magazine training received vehicle following acquisition), to TRAP the neuronal ensemble associated with instrumental acquisition. An additional control group of mice received 4-OHT injection 6 h following the acquisition session ($n = 5$). Mice were the returned to their home cage for 9 days to allow time for virus expression, and then were injected with CNO (3 mg kg$^{-1}$, IP) 15 minutes prior to a retention test to measure the effect of inhibition of this ensemble on lever pressing under extinction conditions (60 min session). Mice were then perfused, and virus expression was visualized and confirmed in the aDLS of 4-OHT treated mice.

*Chemogenetic stimulation experiment.* Fos-Cre$^{ERT2}$ (i.e., FosTRAP1) heterozygous mice were injected with AAV8-hSyn-DIO-hM3D(Gq)-mCherry (M3-DREADD; $n = 8$) into the aDLS and permitted 10 days to recover. Mice underwent 2 consecutive days of magazine training (60 min sessions), and the following day underwent an instrumental acquisition session as described above. Immediately following acquisition, mice received either vehicle ($n = 4$) or 4-OHT injection ($n = 4$), counterbalanced by the latency to acquire 30 pellets. Starting 48 h after the acquisition session, mice underwent daily extinction sessions (60 min) for 8 consecutive days to allow sufficient time for expression of DREADD receptor to occur in TRAPed neurons (10 days from time of 4-OHT injection) and to lower baseline levels of responding such that a stimulatory effect of activating TRAPed neurons could be detected. On the test day (Day 13), mice were injected with CNO (3 mg kg$^{-1}$, IP) 15 min before being placed into the conditioning chamber, and lever-pressing during a retention test under extinction conditions was assessed for 60 min. After the retention test, mice were perfused with 20 mL cold PBS followed by cold 4% PFA, and brains were removed, sectioned on a freezing cryostat, and expression of mCherry was visualized specifically in 4-OHT-treated mice.

**Locomotor activity testing**. To determine if chemogenetic inactivation of the aDLS affects motor behavior, mice expressing hM4Di receptors in the aDLS were placed into an open field arena (Omnitech, Columbus, OH), and allowed to freely explore the apparatus for 30 min. The mice were then removed and injected with vehicle or CNO and returned to the open field arena for an additional 120 min and their locomotor activity recorded.

**Conditioned taste avoidance**. To determine if CNO injection triggered a taste aversion in mice expressing hM4Di receptors in the aDLS similar to the actions of aDLS-infused anisomycin, mice were water-deprived for 2 h then placed into the plexiglass chamber containing a water bottle for 15 min (day 1). The next day (day 2), mice were again water-deprived for 2 h and placed into the plexiglass cage for 15 min, this time containing a bottle that dispensed sucrose solution (20% w/v). Immediately afterwards, mice were injected with vehicle or CNO. On day 3, mice were water-deprived for 2 h and again placed individually into plexiglass cages with a bottle containing sucrose solution (20% w/v) and their sucrose consumption measured.

**Data analyses**. For whole-brain c-Fos analysis, cell count data were normalized by region volume from the Allen Brain Atlas, or from sizes of bins used in custom scripts for striatal subregions, in order to compute regional cell densities. The SciPy library (version 1.4.1) was used to perform all statistics. Cell densities were Z-scored across regions, and the K-Nearest Neighbors and TSNE modules of SciKit-Learn (version 0.22.2) were used to cluster regions. Visualizations were created using Matplotlib (version 3.2.1) and Seaborn (version 0.10.0). All behavioral data were analyzed using GraphPad Prism version 7.0 or later and using custom MATLAB scripts (available from GitHub); see Ref. [67], which were partially adapted from Gritton & Howe et al. [68]. Dendritic spine data were analyzed using GraphPad Prism version 7.0 or later. Calcium imaging data were converted from.avi to multi-page tagged image file (TIF) format, resulting in 9000 frames per imaging session. All image processing and analysis were conducted using custom MATLAB scripts (available from GitHub)[67]. For pre-processing, region of interest (ROI) selection, calcium trace processing, and event characterization, a homomorphic filter was applied to enhance contrast, and motion correction and background subtraction were performed; see Ref. [67]. Manual ROI selection was performed based on morphology using a circle with a radius of 6 pixels from a maximum intensity projection of all frames, and images were smoothed using a Gaussian filter. ROI fluorescence $\Delta F/F$ was calculated as the fluorescence at each time point minus the mean, and then divided by the mean. Threshold for calcium events was set at peak amplitude three standard deviations above baseline and was measured as events per minute. All data are presented as mean ± standard error of the mean (s.e.m.). In all cases, data were analyzed using two-tailed $t$ tests or one- or two-factor analysis of variance (ANOVA) with appropriate between- and within-subject factors. Post-hoc analyses were conducted after statistically significant main effects in ANOVAs. All statistical tests used an α value of $p < 0.05$ for the rejection of the null hypothesis. When appropriate, Grubbs' test was used to identify outliers. Power analysis was conducted with G*Power 3.1.9.6[69].

**Reporting summary**. Further information on research design is available in the Nature Research Reporting Summary linked to this article.

## Data availability
Source data are provided with this paper. All data is available in the main text or the supplementary materials. Related data are available from the corresponding author on reasonable request. No restrictions on data availability apply. Source data are provided with this paper.

## Code availability
Custom code used for ClearMap and calcium imaging analyses is available on GitHub[67].

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

## Acknowledgements

We thank Denise Cai and Tristan Shuman for help with assembly of miniscopes and advice on in vivo calcium imaging. Light-sheet microscopy was performed at the Microscopy CoRE at the Icahn School of Medicine at Mount Sinai; we thank CoRE staff for training and technical support. Figures were created using BioRender.com. This work was funded by grants from the National Institutes of Health (DA043315 and DA007135 to A.C.W.S.; DA025983 to P.J.K.), the Brain and Behavior Research Foundation (NARSAD Young and Distinguished Investigator Grants to A.C.W.S and P.J.K., respectively), iTHRIV Scholars Program (A.G.D.), and a Pfizer Postdoctoral Fellowship Award (R.M.O'.).

## Author contributions

A.C.W.S., S.J., B.J.E. and P.J.K. conceived and designed experiments. A.C.W.S., S.J., R.M.O'C and S.G. performed surgeries, and performed and analyzed behavioral experiments. A.C.W.S. performed iDISCO experiments and analyzed data with the ClearMap package as well as custom Python scripts, available at http://www.github.com/KennyLabSinai/SmithJonkmanNatComm2021[67]. A.C.W.S. and R.M.O'C collected calcium imaging data, and A.G.D. and M.F.R. analyzed these data using custom MATLAB scripts (available from GitHub). A.C.W.S., S.J. and P.J.K. wrote the manuscript with input from all authors.

## Competing interests

The authors declare no competing or conflicting interests related to this work.
