## [Peer Review File · Nature Communications]

Opposing roles for striatonigral and striatopallidal neurons in dorsolateral striatum in consolidating new instrumental actionsREVIEWER COMMENTS

Reviewer #1 (Remarks to the Author):

This manuscript by Smith et al. presents a large and in depth study on striatal function in instrumental learning. The anterior dorsolateral striatum (aDLS) was identified as being crucial for consolidating early instrumental learning based on whole-brain c-Fos analysis and behavioral impairments produced by local protein synthesis inhibition and bidirectional chemogenetic manipulations. Instrumental learning was shown to induce structural plasticity in D1-MSNs, and in vivo calcium imaging detected increased activity in D1-MSNs and decreased activity in D2-MSNs during the post-training consolidation window. Targeted chemogenetic manipulations of D1- and D2-MSNs in aDLS confirmed their opposing roles in consolidating instrumental learning. Interestingly, whereas D2-MSN activity interfered with consolidation of early instrumental learning, it was shown to be required for expression of well-established habitual (devaluation-insensitive) performance. In addition to applying a wide range of powerful and cutting-edge tools to probe neural function, the authors used generally well-controlled and carefully thought out experimental design elements, such as the steps taken to protect against (overshadow) unintended conditioned taste aversion caused by anisomycin. They also incorporated in depth microstructural analysis of instrumental performance, which adds to the richness of the dataset and may provide greater insight into behavioral function. I believe this is a very compelling study that will add much to the field and have only a few minor comments, listed below.

- 1) While I'm generally supportive of lever-press bout analyses, I am not sure they support strong conclusions in the current context. The authors show that blocking protein synthesis in NAc after FR1 training did not significantly alter overall press rates or bout frequency, but reduced bout density/length. Based on this, they claim that "it is likely that the NAc specifically encodes information related to the amount of effort that must be invested in executing a new instrument action as the new action sequence is being encoded." While cited studies show that bout lengths increase with higher ratio requirements, it is not clear how this relates to a situation where only one response is needed for each reward. I think the authors may be tapping into something here but should avoid overstatement. The statement on lines 189-191 is better at acknowledging the suggestive rather than conclusive nature of the finding.
- 2) The calcium imaging data are interesting and generally supportive of the conclusions, but Figure 4i shows that the lack of effect in D1-MSNs for the 10:20 control condition are mostly due to the elevated values in the post-magazine condition, which differs from the 30:0 condition for no clear reason. Is there anything about the nature of these data that helps makes sense of this? Is this just a higher baseline of activity that one would expect to carry forward? Can this be confirmed through other analyses? This is not a major issue, but would be good to address.
- 3) Relatedly, the 10:50 (and 10:20) conditions are interesting because they seem to represent a subthreshold learning, rather than a more obvious no-learning, control. 10:50 animals may not reach the same post-consolidation press rates as fully press-contingent (50:0) groups, but they do seem to show elevated levels relative to noncontingent control (0:50) I would therefore take issue with the framing of the partial training conditions. Receiving a series of noncontingent rewards after earning those reward by lever pressing may also drive new learning, potentially weakening nascent action-outcome contingency learning. This may be relevant to understanding the increase in D2-MSN activity after 10:20 training, which contrasts with the depression after 30:0 training.
- 4) The authors use devaluation testing to confirm the goal-directed (outcome-dependent) vs. habitual (outcome-independent) performance of random interval trained animals. While limited FR1 training is commonly thought to support goal-directed behavior, and there is good reason to question whether such behavior emerges after only a single session of training given recent findings (Iguchi et al. 2017, Scientific Reports). I recommend avoiding direct reference to action-outcome learning (e.g., lines 729, 730, 931, 1060, 1119, 1216) when clear evidence is not provided, using more neutral wording (instrumental learning) when appropriate.

Reviewer #2 (Remarks to the Author):

The authors report findings from a comprehensive set of studies examining the role of striatal sub-

regions and dopaminergic systems in consolidation and expression of a lever press response for food reward in mice and rats. Demonstration of a role for the dorsolateral striatum (DLS) in consolidation of habit memory is consistent with previous studies using various S-R learning tasks. The author's findings provide an extremely interesting and important extension of our understanding of the memory consolidation process in the DLS. Specifically, they provide evidence for the hypothesis that D1 medium spiny neurons (D1 MSNs), and D2 MSNs have differential roles in consolidation of an instrumental response, and the expression of acquired habitual responding, respectively. Interestingly, despite a role in expression of an acquired S-R habit, D2MSN's appear to play an opposing role during initial consolidation. Thus, the findings represent a remarkable and novel dissociation of the role of specific dopaminergic striatal neurons in S-R habit memory consolidation and expression.

The study and conclusions are strengthened considerably by the use of several methodologies and levels of analyses (protein synthesis blockade, c-fos mapping, in vivo calcium imaging, and chemogenetic inhibition). The manner in which each of these methodologies are used and presented to provide converging evidence is impressive. In particular, the use of post-training chemogenetic inhibition to influence memory consolidation in specific neuronal populations is relatively novel, as chemogenetic manipulations are used during "on-line" or behavioral performance in the vast majority of behavioral studies (for a rare example of post-training optogenetic stimulation affecting consolidation see Huff et al., PNAS, 2013). This proof of concept, the utility of using post-training chemogenetic manipulations to examine the role of specific cell types in memory consolidation, is itself an exciting feature of the paper. The behavioral methods and analyses are also sophisticated, including analyses of sub-components of instrumental responding and how they may differ across striatal subregions. Overall, this is an exciting and important set of findings.

I have the following comments for consideration.

1. A minor point concerning the protein synthesis experiment comparing DLS and accumbens. The group n's for the two sites are very different (12 in the accumbens groups, where no effect was observed), and 20 (vehicle) and 30 (drug) in the DLS groups (where an effect was observed). In other learning tasks using post-training manipulations (from inhibitory avoidance to maze tasks), n's much smaller than 30 are the norm. I assume that the large group n's in this task may in part reflect more individual variability in operant lever press behavior? In any case, the huge group difference across the striatal regions does raise the issue of what would happen if the accumbens groups had been doubled to match the DLS groups in size. Nonetheless, this is somewhat minor given the impairing effect of the DLS infusions is consistent with previous studies using other S-R tasks.

2. I was surprised by how much of the Introduction section focused on the nucleus accumbens, rather than literature on the DLS, particularly given the primary findings. The introduction sections detailing inconsistent data/interpretations in previous accumbens studies is much better suited for the Discussion, or at least could be considerably shorter in the intro.

3. The author's definition of "instrumental learning" to open the paper is somewhat awkward. The definition could be reconsidered in a way that will also allow the authors to broaden the impact of the work by finding support from earlier research on the dorsal striatum (including DLS) and habit memory consolidation. The offered definition is that "...instrumental learning occurs when an action produces an unexpected reward...". However, by itself, the occurrence of an unexpected reward activates learning processes across multiple memory systems in the brain, only some of which may involve instrumental responding. Notwithstanding that instrumental learning can obviously also involve punishment, avoidance, and aversive motivation. The term instrumental learning is classically synonymous with operant conditioning, in which Skinner provided a vigorous S-R account of learning. So although we having conditioning 'procedures', the emphasis and more important notion involves the nature of the associations learned. As the authors note, instrumental responding in lever press tasks can involve S-R associations (DLS function), and also other associative structures (e.g. DMS mediated action-outcome associations).

I mention this because previous studies have used tasks involving S-R learning (e.g. maze tasks

for which reinforcer devaluation effects have also been examined), to investigate consolidation processes in various striatal regions, including DLS. These include studies indicate a role for dopaminergic systems as studied in the current paper (e.g. Packard and White, 1991), as well as cholinergic (Prado-Alcala et al; 1977; 1979), and glutamatergic function (Packard and Teather, 1997).

The instrumental/operant response used in the present studies to assess learning can simply be viewed as an exemplar of a task involving acquisition of an S-R association. This view allows for a consistency in interpreting findings across S-R learning in several behavioral settings, and broadens the consideration of impact for the reader (as well as provide additional rationale for the present study). Of course, the dopaminergic activity identified in the present study does not occur in isolation from other dorsal striatal transmitter systems that have been identified as playing a role in habit memory consolidation.

Reviewer #3 (Remarks to the Author):

In this manuscript, Smith and colleagues use operant paradigms, whole brain c-fos mapping, miniscope imaging and chemogenetic manipulations to explore the functional role of the anterior DLS in the consolidation of new instrumental actions. Overall, this is a strong paper, with highly quantitative and rigorous behavioral assays coupled to nicely executed chemogenetic experiments to reveal a new function of anterior dorsolateral striatal subregion. This paper will likely be of interest to many in the field of goal-directed behaviors. Furthermore, there is significant novelty in its careful attention to striatal sub-regions. The overall take home message that aDLS functions in consolidating newly rewarded instrumental actions, in addition to its previously described role in habitual responding, is strongly supported by much of the data. There are a few issues that should be explored further and some that should be relegated to lesser prominence (or strengthened if they need to be kept). The statistical tests are appropriate and there are numerous informative supplemental figures.

Major Points:

x. in Fig1F, given the detailed focus on striatal sub-regions, it would be good to include the other regions (pDLS, aDMS) instead of lumping them together as CPu (if that in fact is what that label represents-somewhat unclear to me).

x. one major point in the manuscript is that the NAc does not in fact mediate encoding on rewarded novel actions patterns. Given that this does conflict existing data, it is noticeable that the same trends exist for NAc (Fig.2I,J) as aDLS (Fig.2d,e), except that it seems to be less powered. This should be addressed.

x. the second half of Fig.3 is an elegant way in which to demonstrate that a subpopulation of recruited aDLS neurons is important for consolidating novel instrumental actions. However, the interpretation is somewhat unclear in its current form and would be improved by: a) use of hM4D instead of excitatory DREADDs which could non-specifically increase operant responding; b) demonstration of a negative control – neurons trapped in non-contingent behavioral scenario (0:50) should not impact subsequent behavior. If this is too much work, supplementing this is also an option.

x. the least clear part of the manuscript is in the description of the activity patterns of SPN subtypes following acquisition. perhaps a physiological as opposed to morphological readout would be more sensitive to bidirectional plastic changes occurring on dSPN/iSPNs? – thinking AMPA/NMDA ratio as a measure of short timescale indicators of plasticity.

x. the in vivo cellular imaging used to get at the patterns of pathway activity after acquisition should be examined further to address a few issues: a) is the appropriate control for analyzing the changes in freq. of events post-magazine training? To me it seems that it is more informative to

compare the post-acquisition in the 10:20 (non-learning) and the 30:0. This would specifically get at differences in activity that related to the consolidation process. As is, the post-magazine comparison would seem to tease out differences in more Pavlovian versus operant behavioral processes. While changing the reference would not affect the iSPN conclusion, it would produce the opposite result for dSPNs (activity goes down during consolidation following learning); b) it seems like an opportunity is missed in the dramatic dimensional reduction of going from spatially unique cellular calcium signals down to event frequency – e.g. does the pattern of SPN activity that goes on during acquisition carry on into the consolidation period? analyses of this type would strengthen the idea that post-learning activity of the population of neurons recruited during acquisition underlies the consolidation process.

x. is the expectation that manipulations in the pDMS would have a similar behavioral effect as the aDLS manipulations? given the c-fos data and pre-existing literature, was pDMS investigated in this behavioral paradigm? this would enhance the generalizability of the results for the field.

Minor Points:

x. is there a good reason for the constant switching back and forth between mouse and rat? would be good to clarify the utility of this approach as well as the spots where observations were made only in one system.

x. it is not clear what exactly is being clustered in Fig1g. How does this really show that plasticity occurs like other plasticity regions? as the main message seems unclear, perhaps this can be supplemented?

x. while there are some attempts at characterizing the site of infection (S.Fig10), this should be done for all viral chemogenetic experiments given the importance of striatal location to the inclusion of these data.

x. it seems likely that D2-Cre mice could label ChINs (unless this is a unique transgenic line), which while not a significant problem for in vivo calcium imaging (relatively small population maybe even distinguishable by soma size), might have larger effects on the observed behavior – this should be discussed.

x. the conclusions about “effort” and NAc anisomycin infusions should be softened in the results and discussion section as this behavior is not explicitly manipulating effort in any measurable way.

x. line 301 should be 5D

Reviewer #1:

Remarks to the author: *This manuscript by Smith et al. presents a large and in depth study on striatal function in instrumental learning. The anterior dorsolateral striatum (aDLS) was identified as being crucial for consolidating early instrumental learning based on whole-brain c-Fos analysis and behavioral impairments produced by local protein synthesis inhibition and bidirectional chemogenetic manipulations. Instrumental learning was shown to induce structural plasticity in D1-MSNs, and in vivo calcium imaging detected increased activity in D1-MSNs and decreased activity in D2-MSNs during the post-training consolidation window. Targeted chemogenetic manipulations of D1- and D2-MSNs in aDLS confirmed their opposing roles in consolidating instrumental learning. Interestingly, whereas D2-MSN activity interfered with consolidation of early instrumental learning, it was shown to be required for expression of well-established habitual (devaluation-insensitive) performance. In addition to applying a wide range of powerful and cutting-edge tools to probe neural function, the authors used generally well-controlled and carefully thought out experimental design elements, such as the steps taken to protect against (overshadow) unintended conditioned taste aversion caused by anisomycin. They also incorporated in depth microstructural analysis of instrumental performance, which adds to the richness of the dataset and may provide greater insight into behavioral function. I believe this is a very compelling study that will add much to the field and have only a few minor comments, listed below.*

Author response: We thank the reviewer for their very supportive comments. As described in detail below, we have generated new data, and revised the manuscript extensively, to fully address each of their concerns. We hope that the reviewer will find our revised manuscript suitable for publication.

Reviewer comment 1: *While I'm generally supportive of lever-press bout analyses, I am not sure they support strong conclusions in the current context. The authors show that blocking protein synthesis in NAc after FR1 training did not significantly alter overall press rates or bout frequency, but reduced bout density/length. Based on this, they claim that "it is likely that the NAc specifically encodes information related to the amount of effort that must be invested in executing a new instrument action as the new action sequence is being encoded." While cited studies show that bout lengths increase with higher ratio requirements, it is not clear how this relates to a situation where only one response is needed for each reward. I think the authors may be tapping into something here but should avoid overstatement. The statement on lines 189-191 is better at acknowledging the suggestive rather than conclusive nature of the finding.*

Author response 1: The reviewer's point is well taken. In response, we have toned down interpretations of our data that relied too heavily on the lever-press bout analyses. However, to address the reviewer's concern more directly, and to provide more support for those bout-dependent interpretations that remain in the revised manuscript, we carried out a new experiment to better understand the behavioral significance of response bouts during the retention test. Specifically, we permitted a group of mice to lever-press during a single training session for chow pellets (FR1-Chow mice), exactly as described in the submitted version of our paper. We permitted a second group of mice to respond for palatable sucrose pellets under a FR1 schedule during the training session (FR1-Sucrose mice). In this manner we could investigate the impact of selectively increasing the magnitude of the earned reward earned during training on subsequent bouts of lever-pressing during the retention test. In parallel, a third group of mice was permitted to respond for chow pellets under a FR2 schedule during the training session (FR2-Chow mice). This allowed us to investigate the impact of selectively increasing the effort required to earn each pellet on subsequent bouts of lever-press during the retention test.

We found that FR1-Sucrose mice emitted approximately twice the number of lever-presses during the retention test as FR1-Chow mice (**Fig. 1**). As FR1-Sucrose and FR1-Chow mice lever-pressed the same number of times during the training session to earn the same numbers of rewards, the increased responding by FR1-Sucrose mice during the retention test reflects the availability of the higher intensity reinforcer available to them during training. The number of lever presses per bout of responding (i.e., bout density) was similar in FR1-Sucrose and FR1-Chow mice during the retention test (**Fig. 1**). By contrast, the number of response bouts (i.e., bout frequency) was much higher in FR1-Sucrose than FR1-Chow mice (**Fig. 1**). Numbers of solitary lever-presses were similar in both groups (**Fig. 1**). These data suggest that bout frequency is more sensitive than bout density to changes in the relative 'value' of the reinforcer earned during new instrumental learning.

FR2-Chow also emitted approximately twice the number of lever presses as FR1-Chow mice during the retention test (**Fig. 1**). This reflects the fact that FR2-Chow mice had to respond twice as frequently as FR1-Chow mice to earn food rewards during the training session. There was a trend for increased bout frequency in FR2-Chow mice compared with FR1-Chow mice during the retention test, which did not achieve statistical significance (**Fig. 1**). However, bout density was markedly increased in FR2-Chow mice compared with FR1-Chow mice (**Fig. 1**). These data suggest that bout density is more sensitive than bout frequency to changes in the effort required to earn a reward during new instrumental learning.

Figure 1. Effects of manipulating reward value and effort during instrumental conditioning on lever-press bout analysis during the retention test. (a) Graphical representation of experimental design. (b) Total number of lever presses in mice during the retention test. (c) Number of response bouts (bout frequency). (d) Presses per response bout (bout density). (e) Solitary lever press responses. * $P < 0.05$, ** $p < 0.01$, post-doc test after significant one-way ANOVA.

These new data support the concept previously reported in the published literature that the propensity of animals to engage in bouts of responding in instrumental tasks is sensitive to the relative value of the reinforcer whereas bout density is sensitive to the amount of effort that must be expended to earn the reinforcer. These new data have been included in the revised manuscript.

Reviewer comment 2: *The calcium imaging data are interesting and generally supportive of the conclusions, but Figure 4i shows that the lack of effect in D1-MSNs for the 10:20 control condition are mostly due to the elevated values in the post-magazine condition, which differs from the 30:0 condition for no clear reason. Is there anything about the nature of these data that helps makes sense of this? Is this just a higher baseline of activity that one would expect to carry forward? Can this be confirmed through other analyses? This is not a major issue, but would be good to address.*

Author response 2: We agree with the reviewer; inspection of the calcium imaging data indeed shows that baseline (post-magazine training) levels of D1 MSN activity were higher in 10:20 mice than 30:0 mice. It is unclear why this should be the case. The 10:20 and 30:0 D1 MSN-GCaMP mice were housed similarly, were surgered and tested on the same days, and both groups had similar handling histories.

The only major difference between the groups was that 10:20 mice earned just 10 reinforcers during the training session whereas 30:0 mice earned 30 reinforcers. However, because baseline D1 MSN activity baseline was recorded before the training session had occurred, this cannot explain the elevated baseline activity in 10:20 mice. It is possible that a stressful event in the vivarium prior to baseline assessment disproportionately impacted the 10:20 mice, resulting in their elevated baseline D1 MSN activity, but we have no evidence that such an event occurred. We apologize that we cannot provide a more satisfactory explanation for this effect.

Reviewer comment 3: *Relatedly, the 10:40 (and 10:20) conditions are interesting because they seem to represent a subthreshold learning, rather than a more obvious no-learning, control. 10:40 animals may not reach the same post-consolidation press rates as fully press-contingent (50:0) groups, but they do seem to show elevated levels relative to noncontingent control (0:50) I would therefore take issue with the framing of the partial training conditions. Receiving a series of noncontingent rewards after earning those reward by lever pressing may also drive new learning, potentially weakening nascent action-outcome contingency learning. This may be relevant to understanding the increase in D2-MSN activity after 10:20 training, which contrasts with the depression after 30:0 training.*

Author response 3: This is an excellent point, and we thank the reviewer for bringing this to our attention. In response, we modified the section of the manuscript that describes the data collected from 10:40 rats to acknowledge that they are unlikely to represent “no-learning” controls but instead are a “subthreshold learning” group, as suggested by the reviewer. Specially, we have now added the following text to the revised manuscript:

“However, the 10:40 rats tended to have higher rates of responding than 0:50 rats during the retention test (Fig. 1b), although this effect failed to reach statistical significance. This suggests that ‘subthreshold’ learning may have occurred in 10:40 rats triggered by successfully earning 10 response-contingent food rewards early in the training session, with this nascent learning weakened by the subsequent withdrawal of the lever and delivery of 40 rewards in a non-contingent manner.”

Second, as pointed out by the reviewer, our calcium imaging data show that D2 MSN activity is increased in 10:20 mice, but decreased in 30:0 mice, during the post-learning consolidation period. Based on the reviewer’s suggestion, we have added the following new text to the revised manuscript to address this interesting observation:

“The fact that 10:20 mice showed increased D2 MSN activity during the post-consolidation period, which is opposite to the decreased D2 MSN activity in 30:0 mice during the same period, suggests that the failure of 10:20 mice to consolidate the new lever-press response may not be a passive process that reflects poor learning because of a limited number of training opportunities. Instead, this may reflect an active process in which D2 MSN activity is engaged during the consolidation phase to ‘overwrite’ a nascent instrumental response that was initially beneficial during the early stages of training session but then rendered obsolete by a change in the instrumental contingencies. Together, these findings suggest that consolidation of a new instrumental action is associated a dramatic shift in the balance of D1 and D2 MSN activity in the aDLS, with increased D1 MSN activity likely involved in consolidating the new response into long-term storage. Conversely, D2 MSNs may execute a ‘quality control’ function, with post-learning decreases in their activity facilitating the consolidation of an advantageous new instrumental response and increases in their activity impeding the consolidation of non-beneficial action sequences.”

Reviewer comment 4: *The authors use devaluation testing to confirm the goal-directed (outcome-dependent) vs. habitual (outcome-independent) performance of random interval trained animals.*

While limited FR1 training is commonly thought to support goal-directed behavior, and there is good reason to question whether such behavior emerges after only a single session of training given recent findings (Iguchi et al. 2017, Scientific Reports). I recommend avoiding direct reference to action-outcome learning (e.g., lines 729, 730, 931, 1060, 1119, 1216) when clear evidence is not provided, using more neutral wording (instrumental learning) when appropriate.

Author response 4: We agree entirely with the reviewer on this point and have now removed any mention of action-outcome learning throughout the revised manuscript.

Reviewer #2:

Remarks to the author: *The authors report findings from a comprehensive set of studies examining the role of striatal sub-regions and dopaminergic systems in consolidation and expression of a lever press response for food reward in mice and rats. Demonstration of a role for the dorsolateral striatum (DLS) in consolidation of habit memory is consistent with previous studies using various S-R learning tasks. The authors' findings provide an extremely interesting and important extension of our understanding of the memory consolidation process in the DLS. Specifically, they provide evidence for the hypothesis that D1 medium spiny neurons (D1 MSNs), and D2 MSNs have differential roles in consolidation of an instrumental response, and the expression of acquired habitual responding, respectively. Interestingly, despite a role in expression of an acquired S-R habit, D2MSNs appear to play an opposing role during initial consolidation. Thus, the findings represent a remarkable and novel dissociation of the role of specific dopaminergic striatal neurons in S-R habit memory consolidation and expression. The study and conclusions are strengthened considerably by the use of several methodologies and levels of analyses (protein synthesis blockade, c-fos mapping, in vivo calcium imaging, and chemogenetic inhibition). The manner in which each of these methodologies are used and presented to provide converging evidence is impressive. In particular, the use of post-training chemogenetic inhibition to influence memory consolidation in specific neuronal populations is relatively novel, as chemogenetic manipulations are used during "on-line" or behavioral performance in the vast majority of behavioral studies (for a rare example of post-training optogenetic stimulation affecting consolidation see Huff et al., PNAS, 2013). This proof of concept, the utility of using post-training chemogenetic manipulations to examine the role of specific cell types in memory consolidation, is itself an exciting feature of the paper. The behavioral methods and analyses are also sophisticated, including analyses of sub-components of instrumental responding and how they may differ across striatal subregions. Overall, this is an exciting and important set of findings.*

Author response: We are grateful to the reviewer for their generous comments and delighted that they consider our new findings important and exciting. We are also very grateful for their thoughtful comments and suggestions, reported below. We have addressed each of these comments/concerns through extensive revisions to the manuscript and/or the addition of new data. We hope that the reviewer will find our revised manuscript suitable for publication.

Reviewer comment 1: *A minor point concerning the protein synthesis experiment comparing DLS and accumbens. The group n's for the two sites are very different (12 in the accumbens groups, where no effect was observed), and 20 (vehicle) and 30 (drug) in the DLS groups (where an effect was observed). In other learning tasks using post-training manipulations (from inhibitory avoidance to maze tasks), n's much smaller than 30 are the norm. I assume that the large group n's in this task may in part reflect more individual variability in operant lever press behavior? In any case, the huge group difference across the striatal regions does raise the issue of what would happen if the accumbens groups had been doubled to match the DLS groups in size. Nonetheless, this is somewhat minor given the impairing effect of the DLS*

infusions is consistent with previous studies using other S-R tasks.

Author response 1: Indeed, there is a large difference in the numbers of animals used for the aDLS-focused anisomycin experiment compared with the other striatal regions that we targeted. The reason for this difference is not because of variation between animals but instead because of our initial skepticism about this finding. This skepticism reflected the fact that the aDLS is generally thought to specialize in stimulus-response learning, which we thought unlikely to be involved in the single-trial learning procedure we used. Instead, our *a priori* hypothesis was that the pDMS regulates the consolidation of new instrumental responses, a hypothesis supported by the results of our whole-brain c-Fos mapping experiment (see Fig. 1 in revised manuscript) and an extensive published literature showing that the pDMS regulates the expression of previously learned instrumental responses. We were therefore surprised that post-learning anisomycin infused into the aDLS but not pDMS disrupted the consolidation of a newly learned instrumental response. However, as noted by the reviewer (see comment #3 below), traditional accounts of instrumental conditioning leaned heavily on stimulus-response learning to explain performance in instrumental tasks, which hinted at the potential involvement of the aDLS. It was based on our initial skepticism about aDLS involvement in the consolidation process that we sought to replicate the finding in two additional cohorts of animals. We consistently found that post-learning anisomycin infused into aDLS disrupted the consolidation process, so we are very confident in the robustness of this finding. That we collapsed data from rats involved in all aDLS experiments into a single figure explains the large numbers of animals in this group.

We also agree with the reviewer that increasing the numbers of animals included in the accumbens group may have resulted in the post-learning anisomycin manipulation achieving statistical significance. Indeed, post-learning NAc anisomycin infusion had a statistically significant effect on 'bout density' in the current group of rats. By increasing the *n*, we would expect that the lower density of bouts would translate into a lower total number of lever-presses. We have included new text in the revised manuscript to acknowledge this caveat.

Reviewer comment 2: *I was surprised by how much of the Introduction section focused on the nucleus accumbens, rather than literature on the DLS, particularly given the primary findings. The introduction sections detailing inconsistent data/interpretations in previous accumbens studies is much better suited for the Discussion, or at least could be considerably shorter in the intro.*

Author response 2: We have heeded the reviewer's advice and moved the accumbens-related text from the Introduction to the Discussion. We also included new text in the Introduction on the role of the dorsolateral striatum in striatal-dependent learning processes.

Reviewer comment 3: *The author's definition of "instrumental learning" to open the paper is somewhat awkward. The definition could be reconsidered in a way that will also allow the authors to broaden the impact of the work by finding support from earlier research on the dorsal striatum (including DLS) and habit memory consolidation. The offered definition is that "instrumental learning occurs when an action produces an unexpected reward". However, by itself, the occurrence of an unexpected reward activates learning processes across multiple memory systems in the brain, only some of which may involve instrumental responding. Notwithstanding that instrumental learning can obviously also involve punishment, avoidance, and aversive motivation. The term instrumental learning is classically synonymous with operant conditioning, in which Skinner provided a vigorous S-R account of learning. So although we having conditioning "procedures", the emphasis and more important notion involves the nature of the associations learned. As the authors note, instrumental responding in lever press tasks can involve S-R associations (DLS function), and also other associative structures (e.g. DMS mediated action-*

outcome associations). I mention this because previous studies have used tasks involving S-R learning (e.g. maze tasks for which reinforcer devaluation effects have also been examined), to investigate consolidation processes in various striatal regions, including DLS. These studies indicate a role for dopaminergic systems as studied in the current paper (e.g. Packard and White, 1991), as well as cholinergic (Prado-Alcala et al; 1977; 1979), and glutamatergic function (Packard and Teather, 1997). The instrumental/operant response used in the present studies to assess learning can simply be viewed as an exemplar of a task involving acquisition of an S-R association. This view allows for a consistency in interpreting findings across S-R learning in several behavioral settings, and broadens the consideration of impact for the reader (as well as provide additional rationale for the present study). Of course, the dopaminergic activity identified in the present study does not occur in isolation from other dorsal striatal transmitter systems that have been identified as playing a role in habit memory consolidation.

Author response 3: In response to the reviewer's helpful comment, we have now re-defined instrumental conditioning in the Introduction. We also modified how we interpreted our findings in the Discussion section to reflect the reviewer's comment. We hope that these modifications fully address the reviewer's concern.

Reviewer #3:

Remarks to the author: *In this manuscript, Smith and colleagues use operant paradigms, whole brain c-fos mapping, miniscope imaging and chemogenetic manipulations to explore the functional role of the anterior DLS in the consolidation of new instrumental actions. Overall, this is a strong paper, with highly quantitative and rigorous behavioral assays coupled to nicely executed chemogenetic experiments to reveal a new function of anterior dorsolateral striatal subregion. This paper will likely be of interest to many in the field of goal-directed behaviors. Furthermore, there is significant novelty in its careful attention to striatal sub-regions. The overall take home message that aDLS functions in consolidating newly rewarded instrumental actions, in addition to its previously described role in habitual responding, is strongly supported by much of the data. There are a few issues that should be explored further and some that should be relegated to lesser prominence (or strengthened if they need to be kept). The statistical tests are appropriate and there are numerous informative supplemental figures.*

Author response: We are very grateful to the reviewer for their generous comments. As described in detail below, we addressed each issue raised by reviewer through revisions to the manuscript and/or the addition of new data. We hope that these modifications have satisfactorily addressed the reviewer's concerns and that our revised paper is now suitable for publication.

Reviewer comment 1: *In Fig1F, given the detailed focus on striatal sub-regions, it would be good to include the other regions (pDLS, aDMS) instead of lumping them together as CPU (if that in fact is what that label represents-somewhat unclear to me).*

Author response 1: We agree with the reviewer on this point. Fig. 1F in the revised manuscript highlights changes in Fos expression in the aDLS, pDMS and NAc, with the bulk 'CPU' data now removed.

Reviewer comment 2: *One major point in the manuscript is that the NAc does not in fact mediate encoding on rewarded novel actions patterns. Given that this does conflict existing data, it is noticeable that the same trends exist for NAc (Fig.2I,J) as aDLS (Fig.2d,e), except that it seems to be less powered. This should be addressed.*

Author response 2: The reviewer makes a good point. In response, we modified the relevant sections of

the manuscript to soften language suggesting that the NAc does not encode reward-related properties of reinforcers but instead encodes only effort-related information during new instrumental learning. We also included the caveat that increasing the numbers of animals in the accumbens-anisomycin experiment may have yielded a statistically significant effect on overall lever-pressing behavior.

Reviewer comment 3: *The second half of Fig.3 is an elegant way in which to demonstrate that a subpopulation of recruited aDLS neurons is important for consolidating novel instrumental actions. However, the interpretation is somewhat unclear in its current form and would be improved by: a) use of hM4D instead of excitatory DREADDs which could non-specifically increase operant responding; b) demonstration of a negative control [for] neurons trapped in non-contingent behavioral scenario (0:50) should not impact subsequent behavior. If this is too much work, supplementing this is also an option.*

Author response 3: We thank the reviewer for these suggestions. We agree that new data showing that chemogenetic inhibition of only those cells active during consolidation (using TRAP) would significantly strengthen our paper. We also agree that the inclusion of additional control groups of mice to confirm the specificity of the TRAP procedure would also strengthen the paper. Therefore, we completed the new experiment suggested by the reviewer. Specifically, we injected AAV-DIO-hM4Di-mCitrine into the DLS of four groups of Fos-TRAP mice. After magazine training, we injected two groups of these mice with either vehicle and or 4-hydroxytamoxifen (4-OHT) to trigger Cre expression and thereby drive hM4Di expression in aDLS neurons active after magazine training. We then trained all mice to lever-press for 30 food pellets during a single training session, and injected the otherwise untreated mice with 4-OHT immediately after the training session or 6 h later. In this manner, we could drive hM4Di expression in aDLS neurons active during consolidation or 6 h later consolidation is insensitive to aDLS manipulations. Finally, we treated all 4 groups of mice with CNO 10 min before a retention test conducted 48 h later.

We found that CNO markedly decreased lever-pressing during the retention test only in Fos-TRAP mice injected with 4-OHT immediately after new instrumental conditioning (Fig. 2). CNO had no effects on retention in the other 3 groups of Fos-TRAP mice (Fig. 2). These exciting new data have been incorporated into the relevant figure in the resubmitted paper (Fig. 3f,g in revised manuscript).

Reviewer comment 4: *The least clear part of the manuscript is in the description of the activity patterns of SPN subtypes following acquisition. perhaps a physiological as opposed to morphological readout would be more sensitive to bidirectional plastic changes occurring on dSPN/iSPNs? ??? thinking AMPA/NMDA ratio as a measure of short timescale indicators of plasticity.*

Author response 4: We apologize that this section of the text was unclear. In response, we modified those sections to improve their clarity. We hope that the revisions have fully addressed the reviewer's concern.

Reviewer comment 5: *The in vivo cellular imaging used to get at the patterns of pathway activity after acquisition should be examined further to address a few issues: a) is the appropriate control for analyzing the changes in freq. of events post-magazine training? To me it seems that it is more informative to compare the post-acquisition in the 10:20 (non-learning) and the 30:0. This would specifically get at differences in activity that related to the consolidation process. As is, the post-magazine comparison would seem to tease out differences in more Pavlovian versus operant behavioral processes. While changing the reference would not affect the iSPN conclusion, it would produce the opposite result for dSPNs (activity goes down during consolidation following learning); b) it seems like an opportunity is missed in the dramatic dimensional reduction of going from spatially unique cellular calcium signals down to event frequency ??? e.g. does the pattern of SPN activity that goes on during acquisition carry on into the consolidation period? analyses of this type would strengthen the idea that post-learning activity of the population of neurons recruited during acquisition underlies the consolidation process.*

Author response 5: Based on the reviewer's suggestion, we compared post-acquisition activity of D1 and D2 MSNs between 10:20 and 30:0 mice. There was no difference in D1 MSN activity between the groups, but there was a striking difference in D2 MSN activity. These findings suggest that post-learning alterations in D2 MSN activity are more robust and perhaps more important than any changes in D1 MSN activity. However, we are reticent to modify the associated Figure in the revised manuscript to reflect these comparisons because, as noted in responses to Reviewer 1 (see comment #2 above), baseline D1 MSN activity measured after the magazine training session was higher in 10:20 than 30:0 mice. We are unsure why this was the case, but it raises the concern that 'between' group comparisons are unreliable unless the data is 'normalized' in some manner to account for this difference in baseline activity. Another concern is the fact that that the 10:20 mice may not be non-learning controls, as we had originally assumed. As pointed out by reviewer 1, the 10:20 mice (and 10:40 rats) appear to have experienced some level of 'sub-threshold' learning. If this, this may also confound 'between group' comparisons. Based on these concerns, we hope that the reviewer will understand our decision to maintain the 'within' group comparisons in the revised manuscript, in which we have greater confidence of robustness and reliability.

The reviewer also encourages us to leverage the power of single cell in vivo calcium imaging technology more fully by assessing changes in activity of individual MSNs across the training period. We agree entirely with the reviewer, and it was for this very purpose that we utilized miniscope technology rather than fiber photometry, which only reports population-level 'bulk' fluorescence. Despite our best efforts, and with help from colleagues who are experts in the use of this technology, we were not able to confidently ascribe an identity to individual cells such that we could conclude that were collecting from the very same cell across all recording sessions. Consequently, we were uncomfortable presenting 'longitudinal' data from the same putative cells across different phases of the learning task and it was for this reason that we had to subject our data to the 'dimensional reduction' commented upon by the reviewer. In future studies we hope to better utilize miniscope technology to monitor D1 and D2 MSNs

across the same learning procedure and better utilize the 'feature rich' data that such single cell analyses are likely to yield.

Reviewer comment 6: *Is the expectation that manipulations in the pDMS would have a similar behavioral effect as the aDLS manipulations? given the c-fos data and pre-existing literature, was pDMS investigated in this behavioral paradigm? this would enhance the generalizability of the results for the field.*

Author response 6: Based on the published literature and our own c-Fos expression data, we anticipated that manipulations of the pDMS would have yielded findings similar to those we collected from the aDLS. However, we found that post-learning infusion of anisomycin into the pDMS had no effects on the consolidation of a newly learned lever-press response. It was for this reason that we focused our attention on the aDLS for the subsequent experiments. While our anisomycin data do not support a role for the pDMS in the consolidation process, it is entirely possible that other experimental approaches (DREADD manipulations etc.) or modifications to the experimental conditions would reveal pDMS contributions to the consolidation of newly learned instrumental responses. To address the reviewer's comment, we have added the following new text to the revised manuscript:

"The pDMS showed increased c-Fos expression after new instrumental learning (Fig. 1f) and is known to regulate the expression of previously learned instrumental actions³³, consistent with a role for the pDMS in consolidating new instrumental actions. We were therefore surprised that post-acquisition infusion of anisomycin into the pDMS did not alter any aspect of responding during the subsequent retention test compared with vehicle-infused rats (**Supplementary Fig. 9**)."

Reviewer comment 7: *Is there a good reason for the constant switching back and forth between mouse and rat? would be good to clarify the utility of this approach as well as the spots where observations were made only in one system.*

Author response 7: We now provide an explanation for why we used rats or mice for each experiment in the appropriate sections of the revised manuscript.

Reviewer comment 8: *It is not clear what exactly is being clustered in Fig1g. How does this really show that plasticity occurs like other plasticity regions? as the main message seems unclear, perhaps this can be supplemented?*

Author response 8: We apologize that we did not present the whole-brain c-Fos analyses in a clear manner. To address this issue, we modified the revised manuscript to better describe the KNN analyses that we performed and the tSNE plots showing these analyses. Specifically, we have added the following text to the relevant section of the manuscript:

"We performed unbiased K-nearest neighbor (KNN) clustering and t-distributed stochastic neighborhood embedding (tSNE) analyses on density of c-Fos+ cells across the entire brains of 0:30 and 30:0 mice (Fig. 1g). The KNN analysis clustered together brain sites in which densities of c-Fos+ cells covaried across animals in the same groups, such that their closer proximity in the tSNE plot reflected greater covariance (Fig. 1g). This revealed that the aDLS and pDMS in 30:0 mice but not in 0:30 mice cluster with brain regions known to regulate learning and memory, such as components of the hippocampal complex and the amygdala (Fig. 1g). Further, KNN analysis of differences in the density of c-Fos expression between mice in the 30:0 and 0:30 groups showed that acquisition of the new lever-press response resulted in the aDLS and pDMS clustering with cortical and subcortical regions known to regulate the

expression of instrumental actions (Fig. 1g), such as the ventral tegmental area (VTA) and substantia nigra (SN). These data suggest that new instrumental learning modifies neural activity in the pDMS, aDLS, and a broader network of cortical, hippocampal and basal ganglia brain regions involved in learning and motivation.”

Reviewer comment 9: *While there are some attempts at characterizing the site of infection (S.Fig10), this should be done for all viral chemogenetic experiments given the importance of striatal location to the inclusion of these data.*

Author response 9: The reviewer makes a very important point about confirming the precise location of intracranial injection sites in our experiments. In fact, we made strenuous efforts to confirm the precise locations of cannula/injector tracts and virus expression for each rat and mouse used in our experiments. Shown in the relevant supplementary figures are the stereotaxic coordinates for each injection and virus expression site for each animal that we used in these our experiments. By presenting the data in stereotaxic coordinates in this manner, we provide quantitative anatomical information for the animals we used. We apologize that we did not better explain our determined efforts to confirm injection sites in the original version of our manuscript. We have now added new text to the revised manuscript better describing our careful confirmation of injection sites in all cases. We hope that this addresses the reviewer’s concern.

Reviewer comment 10: *It seems likely that D2-Cre mice could label ChINs (unless this is a unique transgenic line), which while not a significant problem for in vivo calcium imaging (relatively small population maybe even distinguishable by soma size), might have larger effects on the observed behavior ??? this should be discussed.*

Figure 3. Effects of post-learning chemogenetic inhibition of cholinergic interneurons in aDLS on new consolidation of a new instrumental response. (a) Total number of lever presses in mice during the retention test in mice treated with saline or CNO immediately after training or CNO 6 h after trained (delayed). (b) Number of response bouts. (c) Presses per response bout.

Author response 10: We agree with the reviewer. This is a very important confound. In response, we added new caveats to the revised manuscript acknowledging that cholinergic interneurons may have contributed to the effects seen in D2-Cre mice. As this is such an important point, we directly investigated whether cholinergic neurons in aDLS influence to the consolidation process. Specifically, we injected AAV8-hSyn-DIO-hM4Di-mCherry into the aDLS of ChAT-IRES-Cre knock-in mice, which express Cre recombinase in cholinergic neurons, waited >3 weeks then trained the mice to lever-press for 30 food rewards during a single training session. We found that immediate or delayed (6 h) post-learning injection of CNO did not alter the total number of lever-presses during the subsequent retention test, although there was a non-statistically significant trend for increased frequency of response bouts (Fig. 3). Therefore, we consider it likely that D2 MSNs in the aDLS play a much more prominent role in consolidating newly learned instrumental responses than local cholinergic interneurons. We hope that the clarifications to the manuscript and the inclusion of these new data fully address the reviewer's concern.

Reviewer comment 11: *The conclusions about “effort” and NAc anisomycin infusions should be softened in the results and discussion section as this behavior is not explicitly manipulating effort in any measurable way.*

Author response 11: In response to the reviewer's concern, and to better support our conclusions, we carried out a new experiment to directly investigate the relationship between response bouts, reinforcer magnitude and effort (see same response to Reviewer 1 above). Specifically, we permitted a group of mice to lever-press during a single-session training session for chow pellets (FR1-Chow mice), exactly as described in the submitted version of our paper. We permitted a second group of mice to respond for palatable sucrose pellets under a FR1 schedule during the training session (FR1-Sucrose mice). In this manner, we could determine the effects of selectively increasing the magnitude of the reward earned during training on subsequent bouts of lever-pressing during the retention test. A third group of mice was permitted to respond for chow pellets under a FR2 schedule during training (FR2-Chow mice). This permitted us to explore the effects of selectively increasing the amount of effort required to earn each a reinforcer during training on subsequent bouts of lever-pressing during the retention test.

We found that FR1-Sucrose mice emitted approximately twice the number of lever-presses during the retention test as FR1-Chow mice (Fig. 4). As FR1-Sucrose and FR1-Chow mice lever-pressed the same number of times during the training session to earn the same numbers of rewards, the increased responding by FR1-Sucrose mice during the retention test reflects the availability of the higher intensity reinforcer available to them during training. The number of lever presses per bout of responding (i.e., bout density) was similar in FR1-Sucrose and FR1-Chow mice during the retention test (Fig. 4). By contrast,

Figure 4. Effects of manipulating reward value and effort during instrumental conditioning on lever-press bout analysis during the retention test. (a) Graphical representation of experimental design. (b) Total number of lever presses in mice during the retention test. (c) Number of response bouts (bout frequency). (d) Presses per response bout (bout density). (e) Solitary lever press responses. * $P < 0.05$, ** $p < 0.01$, post-doc test after significant one-way ANOVA.

the number of response bouts (i.e., bout frequency) was much higher in FR1-Sucrose than FR1-Chow mice (**Fig. 4**). Numbers of solitary lever-presses were similar in both groups (**Fig. 4**). These data suggest that bout frequency is more sensitive than bout density to changes in the relative 'value' of the reinforcer earned during new instrumental learning.

FR2-Chow also emitted approximately twice the number of lever presses as FR1-Chow mice during the retention test (**Fig. 4**). This reflects the fact that FR2-Chow mice had to respond twice as frequently as FR1-Chow mice to earn food rewards during the training session. There was a trend for increased bout frequency in FR2-Chow mice compared with FR1-Chow mice during the retention test, which did not achieve statistical significance (**Fig. 4**). However, bout density was markedly increased in FR2-Chow mice compared with FR1-Chow mice (**Fig. 4**). These data suggest that bout density is more sensitive than bout frequency to changes in the effort required to earn a reward during new instrumental learning.

These new data support the concept previously reported in the published literature that the propensity of animals to engage in bouts of responding in instrumental tasks is sensitive to the relative value of the reinforcer whereas bout density is sensitive to the amount of effort that must be expended to earn the reinforcer. These new data have been included in the revised manuscript.

Reviewer comment 12: *line 301 should be 5D*

Author response 12: We thank the reviewer for catching this typo, which has been corrected in the revised manuscript.

REVIEWERS' COMMENTS

Reviewer #1 (Remarks to the Author):

The authors have satisfactorily addressed all of my concerns and were generally very responsive to all critiques.

Reviewer #2 (Remarks to the Author):

authors have adequately addressed reviewer comments/concerns.

Reviewer #3 (Remarks to the Author):

I apologize to the authors and editors for my delayed response. i think this paper is ready for publication. i am particularly impressed with the authors attempts to address all of my concerns - particularly the ones that required more experimental work. i find the experiment showing the specificity of Fos labeling to be incredibly rigorous (and one of the clearest versions of an experiment like this). i also love that they used a ChAT-Cre to further refine their D2-SPN results. this has made a great paper even better.